# Sweetening with Agavins: Its Impact on Sensory Acceptability, Physicochemical Properties, Phenolic Composition and Nutraceutical Potential of Oak Iced Tea

**DOI:** 10.3390/foods14050833

**Published:** 2025-02-28

**Authors:** Aylín Araiza-Alvarado, Saúl Alberto Álvarez, José Alberto Gallegos-Infante, Jorge Alberto Sánchez-Burgos, Nuria Elizabeth Rocha-Guzmán, Silvia Marina González-Herrera, Martha Rocío Moreno-Jiménez, Rubén Francisco González-Laredo, Verónica Cervantes-Cardoza

**Affiliations:** 1Laboratorio Nacional CONAHCYT de Apoyo a la Evaluación de Productos Bióticos (LaNAEPBi), Unidad de Servicio Tecnológico Nacional de México, TecNM-I.T. de Durango, Felipe Pescador 1830, Durango 34080, Mexico; 22041489@itdurango.edu.mx (A.A.-A.); 10040546@itdurango.edu.mx (S.A.Á.); agallegos@itdurango.edu.mx (J.A.G.-I.); sgonzalez@itdurango.edu.mx (S.M.G.-H.); mrmoreno@itdurango.edu.mx (M.R.M.-J.); rubenfgl@itdurango.edu.mx (R.F.G.-L.); veronica.cervantes@itdurango.edu.mx (V.C.-C.); 2Food Research Laboratory, Technological Institute of Tepic, National Technological Institute of Mexico, Instituto Tecnológico Avenue No 2595, Lagos del Country, Tepic 63175, Mexico; jsanchezb@tepic.tecnm.mx

**Keywords:** iced tea, oak, phenol–agavin interaction

## Abstract

Oak infusions enriched with agavins may offer nutraceutical benefits in the development of iced teas. This study evaluated infusions of *Quercus sideroxyla* and *Quercus eduardii* leaves formulated with different concentrations of agavins (0, 2, 6, and 10%), analyzing their physicochemical and sensory properties, chemical stability, and antioxidant capacity. The incorporation of agavins resulted in substantial modifications to physicochemical parameters, including pH, titratable acidity, and soluble solids, thereby enhancing product stability and consistency. Notable distinctions were observed between the two species with respect to their acidogenic response and soluble solid concentration. Interactions between agavins and phenolic compounds, as discerned by UPLC-PDA-ESI-MS/MS and FT-IR, exerted a significant influence on bioactivity of the phenolic constituents, thereby affecting the nutraceutical potential of the infusions. These interactions, facilitated by hydrogen bonds, led to reduction in phenolic acids, such as quinic acid (↓ 43%), and alteration in antioxidant capacity at high concentrations of agavins. The findings underscore the significance of meticulously designing balanced formulations that optimize chemical stability, functionality, and sensory acceptance, thereby ensuring the quality of the final product.

## 1. Introduction

Growing consumer awareness on the correlation between food and health has resulted in a notable surge of interest in iced tea beverages [1,2]. In particular, ready-to-drink teas or iced teas, especially those derived from natural ingredients, are gaining traction due to their perceived health benefits and convenience [3].

Oak leaves, which are by-products of the timber industry, have the potential to offer therapeutic benefits due to their bioactive properties. For example, the potential health benefits of *Quercus sideroxyla* and *Quercus eduardii* have been demonstrated in scientific studies. Infusions prepared from these oak species have been found to contain bioactive polyphenols, including gallic acid, ellagic acid, catechins, and quercetin derivatives [4,5,6,7]. Furthermore, infusions prepared from the leaves of *Q. sideroxyla* and *Q. eduardii* have been demonstrated to exert notable anticancer and anti-inflammatory effects in rat models of colon carcinogenesis, reducing tumor formation and levels of inflammatory markers [8,9]. In vitro studies have demonstrated that *Q. sideroxyla* bark extracts possess hypoglycemic activity and α-amylase inhibitory effects [10]. The antioxidant capacity of infusions and polyphenolic extracts of these oak species has been corroborated by a number of assays [11,12]. Furthermore, proanthocyanidins and other phenolic compounds present in the bark of *Q. sideroxyla* contribute to the species’ notable antioxidant properties [13].

Collective evidence indicates that *Q. sideroxyla* and *Q. eduardii* have the potential to be incorporated into therapeutic formulations, including innovations in herbal health drinks such as iced tea. However, these elements frequently possess flavors that are not appealing to consumers, demanding modifications with the incorporation of additives or sweeteners. In view of the aforementioned, a number of studies have been conducted with the objective of formulating and classifying functional beverages, with a particular focus on those sweetened with plant-based sweeteners. Nanasombat et al. [14] and Carbonell-Capella et al. [15] developed functional beverages with high antioxidant and anti-acetylcholinesterase activity using a variety of plants and sweeteners.

Dube and Ritu [16] and Yadav et al. [17] focused on the utilization of stevia, a natural sweetener derived from *Stevia rebaudiana*, in the formulation of fruit- and whey-based beverages. Likewise, Pawar et al. [18] and Hernández-Pérez et al. [19] provided a broader perspective on the application of plant-derived sweeteners in the food and beverage industry, including biotechnological sweetener production. Furthermore, Hussain et al. [20] and Fawibe et al. [21] analyzed the saccharide and polyol compositions of sweet-tasting plants, highlighting the potential of naturally occurring sweeteners as alternatives to artificial low-calorie sweeteners. However, while stevia and inulin have been extensively studied in various beverage formulations, research on agavins remains limited. Stevia, renowned for its high sweetness intensity, potential bitter aftertaste, and inulin, which primarily functions as a prebiotic fiber with mild sweetness, represents two distinct categories of sweeteners. Agavins, on the other hand, offer a unique combination of prebiotic properties and natural sweeteners without inducing a glycemic response.

Agavins are complex fructan polymers derived from *Agave* plants, which are characterized by their branched structure with β (2–1) and β (2–6) linkages [22]. These polysaccharides exhibit a wide degree of polymerization (DP), ranging from 3 to 60 units [22,23]. The structural complexity of agavins influences their diverse applications in the food, nutraceutical, and pharmaceutical industries [24]. Their potential uses include prebiotics, soluble fibers, stabilizers, and sweeteners [25]. Their structural complexity and interaction with polyphenol-rich matrices, such as oak iced tea, remain largely unexplored. A critical gap in the current research landscape is the understanding of their effects on flavor perception, polyphenol stability, and sensory acceptability in oak-based beverages.

The production of iced tea with leaves of *Q. sideroxyla* and *Q. eduardii*, sweetened with agavins, requires precise management of the techno-functional and physicochemical parameters to maximize the nutraceutical potential of the product. Consequently, it is of the utmost importance to comprehend the chemical interactions between ingredients, guaranteeing that the benefits of bioactive compounds are maintained, while simultaneously ensuring that the stability, taste, and quality of the final product remain uncompromised. Therefore, the objective of this study was to develop and characterize an iced tea made with oak leaves sweetened with agavins, evaluating its sensory acceptability and chemical composition. In addition, a comprehensive analysis was conducted to determine how the interaction between agavins and phenolic constituents of iced tea influences its physicochemical parameters and its nutraceutical potential, considering antioxidant activity and angiotensin-converting enzyme inhibition as responses.

## 2. Materials and Methods

### 2.1. Reagents

The standards for phenolic acids and flavonoids were obtained from Sigma-Aldrich (St. Louis, MO, USA). Phenolic acids encompass a diverse range of compounds, including quinic, caffeic, syringic, chlorogenic, coumaric, ferulic, *trans*-cinnamic, 3,4-di-caffeoylquinic, shikimic, gallic, protocatechuic, and ellagic acids. Flavonoids include catechin, epicatechin, procyanidin B1, quercetin 3-O-β-glucuronide, quercetin 3-O-glucoside, rutin, kaempferol 3-O-glucoside, and phloridzin dihydrate. The methanol utilized for the preparation of the standards was of LC-MS grade quality. Furthermore, the following compounds were utilized: 6-hydroxy-2,5,7,8-tetramethylchroman-2-carboxylic acid (Trolox), 2,2′-azobis (2-amidinopropane) dihydrochloride (AAPH), and 2,4,6-tris (2-pyridyl)-s-triazine (TPTZ). The following reagents were obtained from Sigma-Aldrich: ammonium persulfate, diammonium salt of 2,2′-azino-bis (3-ethylbenzothiazoline-6-sulfonic acid) (ABTS), and fluorescein. The angiotensin-converting enzyme (ACE) inhibitory activity assay was sourced from Dojindo Laboratories (Dojindo Molecular Technologies, Inc., Rockville, MD, USA). Agavins (90% inulin and 10% glucose/fructose/sucrose content) were obtained from American Foods, Guadalajara, Mexico.

### 2.2. Plant Material

The leaves of *Quercus sideroxyla* Bonpl. and *Quercus eduardii* Trel. were collected in July 2021 and subsequently identified and registered with voucher numbers 61,484 and 61,485, respectively, in the herbarium of the CIIDIR-IPN Unidad Durango. The initial step in the procedure involved disinfection of the oak leaves using a 1% solution of sodium hypochlorite for 5 min. This was followed by two rinses with purified water. Subsequently, the oak leaves were subjected to shade drying at 25 °C. Thereafter, the leaves were crushed to a particle size of 2 mm using a DPM-JUNIOR hammer mill. Finally, the leaves were vacuum packed for storage until required.

### 2.3. Iced Tea Formulation

The formulation of the beverages was based on two factors: the species of oak (*Quercus sideroxyla* and *Quercus eduardii*) and the proportion of agavins (0, 2, 6, and 10%). Initially, oak infusions were prepared at 1% (*w*/*v*) in a Royal Prestige brand stainless steel pot equipped with a stainless steel filter. The plant material, with a particle size of 2 mm, was placed in the filter, and water was dispensed. Then, the mixture was heated to boiling temperature and removed from the heat source. Subsequently, the infusions were permitted to stand in the presence of the plant material, thereby facilitating the leaching of soluble compounds for approximately 30 min until they reached room temperature (25 °C). Thereafter, the infusions were sweetened with agavins (2, 6, and 10%). The formulations were bottled in sterile plastic containers under aseptic conditions and stored at −84 °C for the different studies.

### 2.4. Sensory Evaluation

According to the stipulations set forth in the Declaration of Helsinki, all individuals participating in the sensory examinations provided informed consent in the form of a signed document. Moreover, the research protocol was approved by the TecNM/I.T. Durango Scientific Ethics Committee (CEI-003-2022-0301-016).

A free choice profiling (FCP) test was conducted using an untrained panel of 20- to 30-year-old university students to identify the primary sensory attributes of oak iced tea (*n* = 10). The participants were presented with 15 mL samples of the beverage at 4 °C. A quantitative description analysis (QDA) was employed to assess the intensity of the characteristics using a five-point scale, with 1 representing “not at all” and 5 signifying “very intense”. A total of 15 university students, aged 20 to 30 years, participated in the study. For their sensory training, three sessions of 40 min were carried out to identify descriptors in taste to familiarize them with the terminology and use of scales. Each subject was provided with a 15 mL sample of iced tea at 4 °C, with and without agavins. Additionally, a rank ordering test was conducted in which panelists were required to order the samples, with 1 representing the most liked and 4 the least liked. The panel consisted of university students between the ages of 20 and 30 years (*n* = 24). Finally, a focus group test was conducted with a panel of untrained 20- to 30-year-old university students (*n* = 24). The test employed a nine-point scale, with endpoints labeled “9” (indicating a high level of liking) and “1” (indicating a high level of disliking).

### 2.5. Physicochemical Parameters

According to Mexican standards, pH values were obtained using a Hach^®^ potentiometer (model Sen Ion 1, Hach CO, Loveland, CO, USA). The total titratable acidity (TTA) was determined at room temperature (25 °C). To this end, a 5 mL sample of each infusion was diluted in 35 mL of distilled water, and 0.1 N sodium hydroxide was employed. An alcoholic solution of phenolphthalein at 1% was utilized as the indicator. Additionally, soluble solids were quantified with a manual refractometer (model N-1E, ATAGO CO., Minato-ku, Tokyo, Japan), and the results were expressed in °Brix.

### 2.6. Standard Chemical and Phenolic Profile Analysis

#### 2.6.1. Total Phenolic Content

The total phenolic content was determined by the method proposed by Singleton and Rossi [26], with certain modifications. In brief, dilutions of each beverage were prepared in a 1:100 ratio. In a 96-well plate, 25 µL of the diluted sample was added, followed by 80 µL of distilled water and 5 µL of Folin-Ciocalteu reagent (1 N). The mixture was then incubated for 5 min in the dark, after which the oxidation reaction was neutralized with 80 µL of sodium carbonate solution (7.5%). The mixture was then allowed to stand in the dark for 30 min at room temperature, after which the absorbance was measured at 750 nm using a Synergy plate reader (Bio-Tek, Winooski, VT, USA). The total phenolic content was calculated using a calibration curve with gallic acid as the standard, with the results expressed as μg of gallic acid equivalents per mL of the sample.

#### 2.6.2. Total Flavonoids

The total flavonoid content was quantified in accordance with Oomah et al. [27]. A 96-well microplate was prepared with 20 µL of the diluted beverage (1:100) added to each well. Subsequently, 6 µL of a 5% sodium nitrite solution and 12 µL of a 10% aluminum chloride solution were added. Finally, 40 µL of 1 M sodium hydroxide solution and 122 µL of distilled water were added. After allowing the mixture to stand for 5 min at room temperature in the absence of light, a reading at 510 nm was performed using a Synergy plate reader (Waltham, MA, USA). For calculations, a calibration curve was employed, utilizing (+)-catechin as the standard, within a concentration range of 0 and 300 µM. The results were expressed in μg of catechin equivalents per mL of the sample.

#### 2.6.3. Phenolic Profile Analysis via UPLC-PDA-ESI-MS/MS

Standard solutions of flavonoids and simple phenolic acids of HPLC grade were prepared using methanol of MS grade. The beverages were filtered through a 0.45 µm, 13 mm PTFE Acrodisk. This procedure was conducted on a vial with a pre-baked septum. Detection and quantification were accomplished using electrospray ionization/tandem spectrometry in multiple reaction monitoring (MRM) mode. Analysis of the samples (2 µL) was conducted using an ultra-high-resolution chromatography (UPLC) system (Acquity Class-H, Waters Corp., Milford, MA, USA), coupled with a triple quadrupole (Xevo TQ-S, Waters Corp., Milford, MA, USA). The UPLC system included a sample manager maintained at 6 °C and a quaternary solvent manager. A reversed-phase Acquity^®^ BEH C18 column, with a particle size of 1.7 µm and dimensions of 50 mm × 2.1 mm ID (Waters, Corp., Milford, MA, USA), was employed to separate the phenolic compounds. The column was operated at 35 °C.

The elution gradient started at 3% B and subsequently increased to 9% B at 1.23 min. At 3.82 min, the gradient increased further to 16% B and reached 50% B at 11.40 min. By 13.24 min, the gradient returned to 3% B and was maintained for 15 min to allow the column to stabilize. The flow rate was set at 250 µL/min. For the MS/MS assays, multiple reaction monitoring (MRM) was employed. Electrospray ionization (ESI) was performed under negative conditions with the following parameters: capillary voltage of 2.5 kV, desolvation temperature of 400 °C, source temperature of 150 °C, desolvation gas flow of 800 L/h, cone gas flow of 150 L/h, and collision gas flow of 0.13 mL/min. The collision energy was set at 5.0 for the MS mode and 20.0 for the MS/MS mode. A mixture of standards (20 ng/µL) was employed for the purpose of monitoring retention time, m/z values, and MS/MS transitions in the identification of the phenolic profile, which encompasses both phenolic acids and flavonoids (see Appendix A). The chemical characterization of hydrolyzable tannins was performed based on the fragmentation patterns reported by García-Villalba et al. [5] (see Appendix A). The control and data processing of the UPLC and triple quadrupole were conducted using MassLinx v. 4.1 software from the Waters Corporation.

#### 2.6.4. Fourier Transform Spectroscopy in the Infrared (FT-IR)

The freeze-dried formulations of oak iced tea, with and without agavins, were analyzed by FT-IR. The spectra were obtained using a Nicolet iS5 spectrometer (Thermo Fisher Scientific, Tokyo, Japan) at a controlled temperature of 22 °C. Each sample was subjected to 24 scans to optimize the signal-to-noise ratio, with a spectral resolution of 4 cm^−1^. The measurements covered the characteristic spectral region from 4000 to 400 cm^−1^, allowing the identification of functional groups associated with the compounds present in the infusions. Baseline correction was performed using a second-order polynomial fit, ensuring the elimination of possible deviations in absorption due to scattering effects or variations in sample thickness. Spectral deconvolution in the 1200–900 cm^−1^ region was achieved through the implementation of a four-parameter Gaussian fit, employing SigmaPlot v14.0 software (Systat Software, Inc., San Jose, CA, USA). The shape of the bands was modeled as Gaussian, and the baseline was calibrated as a linear function. This analytical approach enabled the enhancement of the resolution of the superimposed peaks and the assessment of specific interaction patterns between the extractable compounds of the oak and the agavins.

### 2.7. Nutraceutical Activity

#### 2.7.1. Inhibition of Angiotensin-Converting Enzyme (ACE) Activity

The methodology was implemented using a Synergy plate reader. The reaction was conducted in a 96-well plate in accordance with the specifications of the ACE Kit-WST colorimetric detection assay (Dojindo Molecular Technologies, Inc., Gaithersburg, MD, USA) for 3-hydroxybutyric acid (3HB) generated from 3-hydroxybutyryl-Gly-Gly-Gly. The experimental procedure entailed mixing 20 µL of each iced tea sample with 20 µL of the substrate buffer in a 96-well plate. Two blanks were prepared in the absence of the sample, one containing the enzyme in the mixture (positive control) and the other without the enzyme (negative control). The plate was incubated at 37 °C for 1 h, after which 200 µL of the indicator working solution was added to the reaction mixture. The microplate was incubated at 25 °C for additional 10 min. Finally, the absorbance reading was recorded at 450 nm. The percentage inhibition of ACE was calculated using the following equation: ACE inhibition (%) = (A_pc_ − A_it_)/(A_pc_ − A_nc_) × 100, where A_pc_ is the absorbance of the positive control reaction mixture, A_nc_ is the absorbance of the negative control reaction mixture, and A_it_ is the absorbance of the iced tea reaction mixture.

#### 2.7.2. ABTS-Cation Radical Scavenging Capacity

The methodology was executed in accordance with Re et al. [28]. Initially, a concentrated solution of ABTS (2,2′-azino-bis (3-ethylbenzothiazoline-6-sulfonic acid) radical was prepared at a molar concentration of 7 mM in a 1:1 ratio with a 2.5 mM ammonium persulfate solution. Subsequently, the solution was incubated at 25 °C for 16 h prior to use. The working solution was obtained by diluting the concentrated ABTS solution in phosphate buffer (50 mM, pH 7.4).

In clear 96-well microplates, 10 µL of the 1:100 *v*/*v* diluted sample was placed, and 190 µL of the ABTS solution was added. The plate was then shaken on an automatic shaker and allowed to stand in the dark for 10 min. The absorbances at 750 nm were subsequently recorded on an Elisa microplate reader (Synergy HT, Microplate Reader from BioTek, Winooski, VT, USA). Concurrently, a fitted line was performed with Trolox at 0 to 200 µM on the same plate. Absorbances of the samples under study were then interpolated, and the results expressed in µg Trolox equivalents per mL (µg TE/mL).

#### 2.7.3. Ferric Reducing Antioxidant Power (FRAP)

The methodology was executed in accordance with Benzie and Strain [29]. The FRAP solution was meticulously prepared using acetate buffer at a concentration of 400 mM and a pH of 3.6. The TPTZ solution was prepared at 30 mM in HCl (40 mM), and subsequently, a solution of ferric chloride hexahydrate (FeCl_3_.6H_2_O) was prepared at 60 mM. The reaction mixture solution was prepared at a ratio of 10:1:1 (*v*/*v*/*v*).

In a transparent 96-well microtiter plate, 20 μL of the 1:100 *v*/*v* diluted sample was mixed with 180 μL of the FRAP solution. The plate was then shaken on an automatic shaker and incubated at room temperature (25 °C) for 20 min. At the end of the incubation period, a microplate reader was used to record the absorbances at 593 nm. Concurrently, a fitting line was built with Trolox at concentrations ranging from 10 to 200 µM, and the absorbances of the samples under investigation were interpolated. The results were expressed in µg Trolox equivalents per mL (µg TE/mL).

#### 2.7.4. Oxygen Radical Scavenging Capacity (ORAC)

The methodology for measuring ORAC was performed according to Ou et al. [30]. A solution was prepared with fluorescein at 1.09 µM in phosphate-buffered saline (PBS 7.5 mM) and a solution of AAPH (2,2-azobis-2-amidinopropane-dichloro-hydrochloride) at 700 mM.

In a black 96-well plate, 20 μL of the 1:100 *v*/*v* diluted sample was mixed with 200 μL of the 1.09 μM fluorescein solution. The plate was then shaken on an automatic shaker and incubated at 37 °C for 15 min. At the end of the incubation period, 75 µL of the AAPH solution was added to the reaction. Subsequently, the plate was placed in a microplate reader, and the absorbances were recorded at excitation and emission wavelengths of 485 and 590 nm, respectively. The kinetic readings were collected at 210 s intervals for 3 h. Concurrently, a fitting line was built with Trolox at concentrations ranging from 10 to 50 µM. At the end of the process, the areas under the curve were calculated for each standard and sample concentration. PBS was utilized as a blank, and the fitted line was constructed using the areas under the curve resulting from the kinetics performed with the standard. Finally, the values of areas under the curve of samples studied were interpolated, and the results expressed in µg Trolox equivalent per mL (µg TE/mL).

### 2.8. Statistical Analysis

Statistical analysis of the data from the study was performed using several tests to evaluate significant differences and correlations between the parameters studied. First, post hoc Least Significant Difference (LSD) tests with a significance level of *p* < 0.05 were used to identify the differences between the means of the groups evaluated. Pearson’s correlation analysis was employed to assess the relationships between the responses in the physicochemical parameters and the associated variables. Subsequently, a partial least squares discriminant analysis (PLS-DA) was implemented to ascertain the primary phenolic constituents responsible for the observed disparities between iced teas. All statistical analyses were performed using SPSS 25.0 statistical software (IBM Corporation, Armonk, NY, USA).

## 3. Results

### 3.1. Sensory Evaluation

The panelists were able to discern and identify distinctive attributes in both iced tea formulations prepared with leaves of *Q. sideroxyla* and *Q. eduardii*. In the case of *Q. sideroxyla*, the most salient attributes were identified as astringent, bitter, refreshing, and herbal. In contrast, *Q. eduardii* was described as having astringent, refreshing, herbal, and pleasant attributes. These descriptions indicate a clear differentiation between the two species in terms of sensory characteristics, despite some similarities in the descriptors used, such as “astringent” and “refreshing”. This reflects the idiosyncratic perceptions of the panelists, demonstrating how different species can evoke similar attributes, yet exhibit distinctive profiles. In beverages prepared with 10% agavins, sweetened *Q. sideroxyla* exhibited the primary attributes of sweetness, bitterness, astringency, floral notes, and strength, whereas sweetened *Q. eduardii* beverages displayed the attributes of sweetness, blandness, lightness, refreshment, and pleasantness.

To ascertain an objective definition of the intensity of each characteristic, a quantitative description analysis was conducted on beverages sweetened with agave. This analysis quantifies flavor perception at varying agavin concentrations. The results indicate that at agavin concentrations of 6% or greater, sweet and bitter flavors are predominant in *Q. sideroxyla*, whereas in *Q. eduardii*, sweet and bland flavors are prevalent, followed by astringent and bitter flavors. This illustrates the significant impact of natural sweeteners on sensory characteristics, contingent on concentration.

To compare the beverages according to the perceived intensity of one or more attributes, the sensory rank-ordering analysis of the different agavin concentrations (especially at 6 and 10%) confirmed that at higher agavin concentrations, the sweet taste predominates in both species, but with notable differences in the tasteless attribute, which appears stronger in *Q. eduardii*. The test facilitated the ordering and prioritization of the dominant flavors according to the concentration of the agavins utilized. The focus group test demonstrated that agavins are perceived as an acceptable and natural sweetener at concentrations of 6% or higher in *Q. sideroxyla* (Figure 1). Statistical analysis using ANOVA followed by an LSD post hoc test indicated a significant difference at the 6% agavin concentration (*p* < 0.001), confirming its impact on sensory acceptability. The participants indicated that at these concentrations, the iced tea is sufficiently sweet without being excessive. Additionally, they noted a preference for the *Q. eduardii* profile, citing its lighter and more pleasant taste, and observed a diminished perception of the taste identified as bland. The comments from the participants suggest that agavins are an effective sweetener, providing a balance between sweetness and the perception of bland and refreshing flavors. Notwithstanding the aforementioned responses, the sensory acceptability ascribed by the volunteer participants indicates that agavins exerted a considerable impact on the acceptability of beverages formulated with *Q. sideroxyla*, whereas agavin sweetening exerted no discernible influence on *Q. eduardii* beverages. This information is crucial for the development of products focused on the final consumer, as it validates the preferences and level of acceptance of the final product.

### 3.2. Physicochemical Properties

The objective of this study was to assess the impact of agavin addition on the physicochemical properties of oak iced teas, with a focus on pH, titratable acidity, and soluble solids. pH is a crucial parameter for the chemical stability and sensory attributes of beverages; however, in this study, it did not significantly affect consumer acceptability. The incorporation of agavins into *Q. sideroxyla* iced tea led to a statistically significant decrease in pH of 0.1 and 0.2 points at concentrations of 6 and 10%, respectively. In *Q. eduardii*, a more pronounced decrease in pH (0.5 points) was observed at 2% agavins, though this effect was less pronounced at higher concentrations (6 and 10%), while remaining statistically significant (see Table 1).

Titratable acidity, which measures the total acid content, exhibited a twofold increase in *Q. sideroxyla* iced teas with 2% agavins, reaching 0.02 g citric acid/L. In *Q. eduardii*, acidity changes were less pronounced, even at 10% agavins. A significant acidogenic response (*p* < 0.05) was observed in *Q. sideroxyla*, correlating with a decline in pH. However, variations in *Q. eduardii* may be attributable to other constituents. Soluble solids increased with agavin concentration, reaching a peak of 9.3 °Brix in oak iced teas with the highest agavin levels, thereby enhancing sweetness perception and body.

A Pearson correlation analysis was performed to evaluate the relationship between pH, titratable acidity, and °Brix as a function of agavin concentrations. The analysis revealed a negative correlation between agavin concentration and pH, irrespective of the oak species (Table 2). However, a separate analysis was conducted for *Q. sideroxyla* and *Q. eduardii*, as the beverages exhibited divergent responses to the observed correlation between agavin concentration and pH. In both species, a negative correlation between pH and titratable acidity was observed, though this correlation did not reach statistical significance when analyzed separately. However, when the data was analyzed without blocking by species, a significant negative correlation was identified (*p* < 0.05). The °Brix values showed a positive correlation with the total acidity values, regardless of the oak species. The detailed values are presented in Table 2.

### 3.3. Chemical Composition

The pH level affects the interactions between components, such as polysaccharides and polyphenols, which are of great importance in products containing significant amounts of bioactive compounds, such as oak infusions. The statistical analysis, which employed Pearson’s correlation as a distance measure to examine the relationship between standard chemical analyses and physicochemical parameters, revealed that the incorporation of agavins markedly reduces the total phenolic content (*p* < 0.05) in the beverages (see Table 3). This decrease is not correlated with pH values in *Q. sideroxyla* iced teas (R = 0.4096), in contrast to the positive correlation observed in *Q. eduardii* beverages (R = 0.8024). These findings differ from those observed in the correlation analysis of titratable acidity, where a strong negative correlation was identified with the values recorded for *Q. sideroxyla* (R = 0.8821), and no correlation was observed with iced tea made with *Q. eduardii* (R = 0.1514).

In order to evaluate the impact of sweetening with agavins in iced teas, the phenolic profile was explored by mass spectrometry (see Table 4). A total of 23 phenolic acids, 12 flavonoids, and 7 hydrolyzable tannins were identified in the beverages. The concentration of agavins was demonstrated to exert a significant influence on the phenolic contents and profiles (see Figure 2a).

Partial least squares discriminant analysis (PLS-DA) enabled the most significant chemical constituents to be identified for the purpose of distinguishing between samples based on the predictor variables. The findings suggest that agavins with a degree of polymerization greater than 10 at varying concentrations exhibited distinct grouping in the component space (see Figure 2b), indicating that the concentrations of phenolic compounds vary significantly depending on the concentration.

The variance explained by the model was 65.0% in the first two principal components of the iced teas prepared with *Q. sideroxyla*, while for those prepared with *Q. eduardii,* it was 51.7%, indicating that the model was adequate for discriminating between the different experimental conditions. This finding was further substantiated by the R^2^ and Q^2^ descriptors of the model, which exhibited values of 0.9981 and 0.9527 for *Q. sideroxyla*, and 0.9968 and 0.9179 for *Q. eduardii*, respectively. These values substantiate the model’s capacity for prediction.

The analysis revealed that in the iced teas made with *Q. sideroxyla*, the key variables with values greater than 1 were quinic acid, galloyl hexoside (II), tartaric coumaric acid, galloyl quinic acid (I), and 2-hydroxybenzoic acid, among other phenolic acids. These variables decreased drastically in concentration as the agavin content decreased beverage pH values.

In *Q. eduardii*, the significant variables with values greater than 1 were quinic acid and hydroxybenzoic acids, including 2-hydroxybenzoic and 4-hydroxybenzoic acids, as well as a decrease in the flavonoid rutin (see Figure 2c). This decline may be attributable to the interaction of the agavins with these phenolic acids, which could diminish their bioavailability or induce their precipitation through the formation of complexes with the agavins. This effect was not observed in the formulation of beverages. The addition of agavins can modify the physicochemical environment and interact with these phenolic acids and flavonols, thereby enhancing their stability within the system. In both species, an increase in caffeoylquinic acids, such as chlorogenic and cryptochlorogenic acids, and flavonoids, such as procyanidin B2 and quercetin glycoside, was observed. These compounds appear to exhibit greater stability as the concentration of agavins increases.

It is hypothesized that the interaction between the constituents of iced tea from the two oak species and the varying concentrations of agavins could create an environment that favors the stability of these compounds. It is noteworthy that when analyzing agavin solutions at different concentrations, the presence of these phenolic compounds was not identified in the systems.

### 3.4. Physicochemical Properties in the Conjugation of Phenolic Compounds with Agavins

The use of FT-IR monitoring offered a non-destructive and expeditious approach to the examination of chemical interactions between phenolic compounds and agavins. Through the investigation of the alterations in phenolic compounds that were identified as significant variables with values greater than 1 for *Q. sideroxyla*, variations were discerned in specific absorption regions of the infrared spectrum (see Appendix A). As illustrated in Figure 3, the most significant regions within the FT-IR spectrum for both *Q. sideroxyla* and *Q. eduardii* were identified, highlighting the regions where substantial interactions between agavins and phenolic acids were detected. Specifically, the spectrum exhibited agavin bands (1200–900 cm^−1^) and regions of the aromatic groups of phenolic acids, including aromatic C = C (1600–1400 cm^−1^) and phenolic C–OH (1300–1000 cm^−1^) stretchings. Additionally, hydrogen bond regions (3600–3100 cm^−1^) were identified.

A decline in relative intensity was observed in this region with increasing agavin concentrations (6 and 10%), suggesting interactions between agavins and constituents of the infusion, such as quinic acid and hydroxybenzoic acids (see Figure 4). These interactions may involve the formation of hydrogen bonds between the O-H groups of the agavins and the hydroxyl/carboxyl groups of these analytes.

The formation of complexes between agavins and phenolic compounds has been shown to alter the characteristic vibrations of the C-O bond in carbohydrates and decrease the signal in this region. In addition, an increase in the intensity of the bands in this region was observed for the formulation with 2% agavins, which could be associated with a greater availability of specific molecular interactions. This includes an adequate proportion of functional groups available to interact with the phenolic compounds of *Q. sideroxyla*. This interaction can be further enhanced by the presence of hydrogen bonds between the hydroxyl groups of polysaccharides and the carboxyl groups of phenolic acids, such as hydroxybenzoic acids. Additionally, the formation of complex networks through intermolecular interactions can lead to an increase in electronic density and, consequently, band intensity.

In the aromatic region (1600–1400 cm^−1^), alterations in the shape and intensity of the bands associated with the C=C of aromatic rings present in the phenolic acids (2-hydroxybenzoic acid and 2,5-dihydroxybenzoic acid) were identified. These findings suggest the formation of molecular interactions between the aromatic rings and the agavins, which cause a disturbance in the vibrations of the C=C bonds. These interactions may be characterized as either weak π-π stacking between the aromatic rings and the polysaccharide structure, or hydrogen bonds that limit the vibrational freedom of the aromatic groups.

In the region of hydrogen bonds (3600–3100 cm^−1^), the bands corresponding to O-H stretching, associated with free hydroxyl groups and hydrogen bonds, broaden and show a decrease in intensity as the concentration of agavins increases (6 and 10%). The decline in signal intensity observed in this region at these agavin concentrations indicates the formation of hydrogen bonds between the hydroxyl groups of the agavins and the phenolic acids. This phenomenon is postulated to result in a reduction in the number of free O-H groups, thereby leading to a decrease in the intensity of the signal. This phenomenon is consistent with the decrease in the aforementioned phenolic analytes observed in liquid chromatography with mass tandem, as the phenolic acids could be “sequestered” by the agavins through hydrogen bonds, thereby reducing their availability for detection in mass analysis. Consequently, in the formulations of *Q. sideroxyla*, the decrease in the regions 1200–900 cm^−1^, 1600–1400 cm^−1^, and 3600–3100 cm^−1^ suggests that the agavins interact with the phenolic acids (quinic acid and hydroxybenzoic acids) through hydrogen bonds in the region of 3600–3100 cm^−1^. These findings also suggest potential interactions with aromatic groups (1600–1400 cm^−1^) and alterations in the vibrational structure of carbohydrates (1200–900 cm^−1^). Consequently, the reduction of quinic acid and hydroxybenzoic acids in the UPLC-PDA-ESI-MS/MS analysis can be attributed to the formation of non-covalent complexes between the phenolic compounds and the agavins. These interactions have been shown to limit the availability of phenolic acids in solution, thereby making it challenging to detect them in liquid chromatography with mass tandem.

Specifically, the increase in the concentration of agavins in the formulations of *Q. sideroxyla* induces interactions with phenolic acids, evidenced in FT-IR as a decrease in the signals of the key regions (1200–900 cm^−1^, 1600–1400 cm^−1^ and 3600–3100 cm^−1^). These interactions, predominantly through hydrogen bonds and potential structural effects, may elucidate the reduction of quinic acid and hydroxybenzoic acids, as detected by UPLC-PDA-ESI-MS/MS. This may also reflect the complexity of conjugations between polysaccharides, phenolic compounds, and acid-base characteristics. Consequently, phenolic acids, which were identified as significant variables in the chemometric analysis, exhibited a Pearson correlation value of 0.951 with respect to the recorded pH values. This finding suggests that the decrease in free quinic acid is associated with the release of hydrogen ions, which is reflected in the decrease in pH and in a strong negative correlation (R = −0.725) with the values of titratable acidity recorded for the iced teas developed with *Q. sideroxyla* (see Appendix A). Concurrently, the correlations for 2-hydroxybenzoic acid and 2,5-hydroxybenzoic acid were 0.876 and 0.731, respectively, with the recorded pH values. For titratable acidity, the correlations were −0.641 and −0.776, respectively.

In consideration of the impact of agavins incorporated into the infusion of *Q. eduardii* on the production of iced teas, Figure 3 illustrates the predominant key regions within the FT-IR spectrum of *Q. eduardii*. As observed with the formulations with *Q. sideroxyla*, three key regions were identified at 1200–900 cm^−1^, 1600–1400 cm^−1^, and 3600–3100 cm^−1^.

An analysis of the range from 1200 to 900 cm^−1^ (see Figure 5) revealed vibrations primarily associated with functional groups of polysaccharides, such as agavins (C-O, C-C, and sugar ring vibrations). It was observed that the signals undergo changes with increasing percentages of agavins, suggesting the possibility of chemical interactions. A particularly salient increase in the intensity of the bands in this region is observed for the formulation with 2% agavins, in comparison to that observed in the formulation of *Q. sideroxyla*. This effect may be associated with an increased availability of specific molecular interactions, including an adequate proportion of functional groups available to interact with the phenolic compounds of *Q. eduardii*. This effect may maximize hydrogen-bonding interactions between hydroxyl (-OH) groups of polysaccharides and carboxyl (-COOH) groups or other phenolic acids, such as hydroxybenzoic acids. In addition, complex networks can be formed through intermolecular interactions that increase electron density, thereby enhancing the intensity of the bands and the presence of overlaps.

In the analysis with a four-parameter Gaussian fit, an adequate fit was identified for *Q. sideroxyla* (R^2^ = 0.9050) and *Q. eduardii* (R^2^ = 0.9015). However, in the formulations of *Q. eduardii*, it was observed that the region between 978.77 and 1042.13 cm^−^^1^ falls outside the prediction limits of the model for the condition with added 2% agavins. In this region, as previously mentioned, a concentration and molecular dispersion effect may occur, particularly at the 2% concentration for *Q. eduardii*. This is because agavins may be more dispersed due to the pH of the system (5.33, see Table 1), which facilitates greater interaction with phenolic acids and avoids excessive self-aggregation that occurs at higher concentrations (6 or 10%), where the agavin molecules tend to form aggregates. It has been shown that this can induce the interaction of phenols and decrease the intensity, as seen in Figure 5. Furthermore, spectral overlap may occur at low agavin concentrations for *Q. eduardii*, since, at this concentration, the characteristic signals of polysaccharides (C–O, C–C) in this region may coincide more clearly with the phenolic acid bands. This results in an increase in apparent absorbance due to the combined contribution of these signals.

A chemometric analysis was performed, and its results were related to the FT-IR results. It was observed that a relevant region was the 1610–1580 cm^−1^, which is associated with C = C vibrations of aromatic rings and asymmetric stretching of carboxylic groups. Notably, the projection of the chemical analysis exhibited an augmentation in quercetin 3-O-β-glucuronide and procyanidin B1, two flavonoids with strong aromatic character, which have the capacity to enhance the intensity of the bands within this specific region. Conversely, the decline in hydroxybenzoic acids, such as 2-hydroxybenzoic acid and 4-hydroxybenzoic acid, diminished the contribution of vibrations associated with -COOH bonds within the 1610–1580 cm^−1^ region.

Notably, there were increases observed in caffeoylquinic acids, such as chlorogenic and cryptochlorogenic acid, which contain an ester group that is expected to intensify in the region of 1730–1700 cm^−^^1^. Consequently, it was hypothesized that other phenolic compounds, such as the hydrolyzable tannin trigalloyl glucoside, might also contribute to this band due to the presence of ester bonds in their structure. However, the opposite behavior was observed, which explains why these stable metabolites are present in the beverages, possibly due to the acidogenic responses promoted by the addition of agavins, which stabilized them in the system (see Appendix A).

In summary, the interaction between agavins and phenolic compounds has the potential to influence the availability and reactivity of these compounds. The results presented in this section indicate that these interactions may result in a positive effect, which could be characterized as stabilization, or a negative effect, which could be characterized as a decrease in reactivity. Therefore, it is imperative to ascertain the impact of the observed conjugations on the antioxidant activity of the formulations.

### 3.5. Relationship Between Chemical Shifts and Antioxidant Capacity

The findings, as indicated by both FT-IR and UPLC-PDA-ESI-MS/MS analysis, demonstrate that an increase in agavin concentration exerts an influence on the chemical interaction with phenolic compounds, including quinic acid and hydroxybenzoic acids. This, in turn, leads to a corresponding alteration in their antioxidant activity.

These alterations were found to be associated with the discernment of pivotal characteristics through the implementation of PLS-DA analysis (Figure 2). This observation indicates that the interplay between agavins and phenols plays a pivotal role in the modulation of these responses through diverse mechanisms in *Q. sideroxyla* and *Q. eduardii* formulations (see Table 5).

The incorporation of agavins into the formulation has been shown to enhance the inhibitory capacity of its constituents on the angiotensin-converting enzyme (ACE). However, antioxidant response analysis suggests that the conjugation of agavins with phenolic compounds impedes the interaction of crucial antioxidants. Specifically, the analysis of antioxidant capacity by the ABTS, FRAP, and ORAC methods revealed a non-linear response upon the addition of agavins.

At intermediate concentrations of agavins, an enhancement in antioxidant activity is evident, presumably attributable to the stabilization of phenolic compounds, including chlorogenic acids, catechin, procyanidin B1, and quercetin 3-O-β-glucuronic acid. However, at higher concentrations, the effect is diminished or even reversed, likely due to the formation of less active complexes between agavins and phenolic compounds.

The cubic fitting equations presented in Table 6 facilitate the modeling of the effect of agavins and their interactions with polyphenols on antioxidant capacity. These equations demonstrate complex behaviors that are contingent on concentration and specific interactions between compounds.

The fitted cubic models delineated the antioxidant response as a function of agavins (X) concentration, evaluated by ABTS, FRAP, and ORAC methods, and their interactions with phenolic compounds. These models enabled the discernment of intricate trends that reflected the direct impact of agavins on antioxidant activity at specific concentrations. These responses were associated with the stabilization of phenolic compounds, including chlorogenic acids, catechin, and hydrolyzable tannins.

The scavenging capacity of free radicals, such as the cation radical ABTS, exhibited predominantly negative behavior with increasing agavins in *Q. sideroxyla*, but this response was moderated by complex interactions in other species. This stands in contrast to the observations made in the formulations of *Q. eduardii*, where an initial positive response was recorded (+3.84 X), which was subsequently mitigated by negative effects at intermediate concentrations (−1.00 X^2^). When analyzing the response of the formulations to the reducing power of their constituents, a notable improvement was identified in *Q. sideroxyla* with initial increases (+219.88 X). However, the effect decreased at high concentrations (−55.12 X^2^). Conversely, in *Q. eduardii*, an inverse trend was observed, exhibiting adverse effects across the concentration range (−48.52 X). The investigation of the constituents’ effect on oxygen atom absorption revealed an initial positive response (+27.09 X) with slight stabilization at higher concentrations in *Q. sideroxyla* formulations, while in *Q. eduardii* formulations, a more negative behavior was observed (−48.52 X). This finding suggests that agavins may have a reduced capacity to enhance this response in this species.

The equations reflect how agavins modulate antioxidant activity in a nonlinear manner, highlighting the significant contribution of interactions with polyphenols. The goodness of fit (R^2^~0.95–0.98) validates the proposed models and enables the evaluation of the direct and indirect effects of chemical interactions.

In summary, PLS-DA analysis (see Figure 2) demonstrates the impact of varying agavin concentrations on key variables associated with phenolic compounds and antioxidant capacity. The groupings observed in Figure 2 substantiate that interactions between agavins and polyphenols substantially modify chemical profiles, which is congruent with the alterations detected in FT-IR and the decreases identified by UPLC-PDA-ESI-MS/MS. The cubic models in Table 5 further substantiate this relationship by capturing the intricate effects of the interaction between agavins and polyphenols on antioxidant capacity. The relative weights of the quadratic and cubic terms indicate that these interactions are nonlinear, particularly in the case of FRAP for *Q. sideroxyla* (+219.88 X, −55.12 X^2^) and ORAC for *Q. eduardii* (−44.46 X, +12.56 X^2^). The correlation between the ABTS, FRAP, and ORAC models demonstrates consistent trends, thereby suggesting that chemical interactions uniformly affect the antioxidant mechanisms of both species.

## 4. Discussion

The utilization of agavins as sweeteners in many food products has yielded promising outcomes. Nevertheless, there is a paucity of literature that assesses their capacity to sweeten beverages. Agavins have been demonstrated to enhance the sensory characteristics of food products, including yogurt [31] and ice cream [32], by improving mouthfeel and texture. The incorporation of agavins into mozzarella cheese has been demonstrated to enhance the fiber content and texture, while maintaining consumer acceptance [33]. The incorporation of agave fructans into a powdered smoothie resulted in favorable sensory attributes and a moderate glycemic index [34]. The extraction of agavins from *Agave durangensis* leaves demonstrated their potential for incorporation into food products [35]. Overall, agavins become a promising sugar substitute as they improve sensory properties. However, in conventional industrial processes involving the pasteurization of beverages to guarantee their microbial safety, it is possible that the structural stability of agavins could be affected. Studies by Muñiz-Márquez et al. [36] have demonstrated that the application of heat to agave syrup (121 °C, 15 min) results in the depolymerization of fructans, leading to a reduction in fructooligosaccharides, such as kestose, and an increase in simple sugars containing glucose, fructose, and sucrose. Consequently, the objective of formulating iced tea made from oak as a premium drink is to preserve its constituents; nevertheless, further investigation is required to ascertain their effect on different techno-functional aspects in the development of pasteurized iced tea.

The study demonstrates that the addition of agavins has a notable influence on the physicochemical parameters of oak infusions. The impact on pH and interactions between polysaccharides and bioactive compounds is necessary, as these parameters are critical to ensure long-term sensory and chemical stability of the product. The pH range of oak infusions is typically between 5.0 and 6.0 [6,37,38]. In the present study, the initial pH values of the unsweetened *Q. sideroxyla* and *Q. eduardii* infusions were 5.42 and 5.84, respectively. The influence of agavins on the pH of *Q. sideroxyla* was found to be significant (*p* < 0.001), though the overall decrease (up to 0.2 points at 10%) remained modest. In contrast to the reduction in pH and increase in acidity commonly observed in commercially prepared infusions [39], no correlations were detected in these beverages between pH and the significant increases in acidity in *Q. sideroxyla* and *Q. eduardii* infusions resulting from the addition of agavins (see Table 2). A notable observation in the formulation of *Q. eduardii* was the decrease in pH that occurred with the addition of 2% agavins (5.84 to 5.33), followed by an increase with the addition of 6% (5.70) and a decrease with the addition of 10% (5.62). This phenomenon may be attributed to acid–base interactions between the agavins, and the polyphenols in the oak leaves. This phenomenon could be attributed to the presence of functional groups capable of forming hydrogen bonds between the hydroxyl groups of polysaccharides and the carboxyl groups of phenolic acids, such as hydroxybenzoic acids. The fructan structure of agavins has been hypothesized to enhance the solubility and dissociation of phenolic acids, thereby affecting pH levels. The observed non-linear response may be attributable to competitive interactions that vary with concentration. The polyphenolic profile of *Q. eduardii* and *Q. sideroxyla* could also play a role, suggesting the need for further research.

Ready-to-drink infusions usually have lower pH values and higher titratable acidity values than freshly prepared infusions, suggesting a possible erosive effect [39].

The pH of oak infusions sweetened with agavins was found to be higher than the typical range of commercial iced tea (3.5–4.5), which is relevant to their sensory perception. In comparison, sucrose tends to maintain a neutral or slightly acidic pH, depending on the conditions of herbal infusion, while stevia has a minimal impact on this parameter, and its perceived sweetness is significantly different. In terms of °Brix and titratable acidity, the agavin formulations presented values comparable to those of other functional beverages sweetened with honey or natural syrups, which suggests their viability in the premium beverage industry.

However, a drastic increase in titratable acidity was not observed in our study. This demonstrates that the addition of agavins has a notable influence on the physicochemical parameters of the oak infusions. At concentrations of 2% or greater, agavins have been observed to double the acidity of *Q. sideroxyla* (0.02 g CA/L) and cause a significant increase in soluble solids in both species. These findings suggest a positive impact on the perception of sweetness and body of the beverage. However, further monitoring of the impact on pH and interactions between polysaccharides and bioactive compounds is necessary, as these parameters are critical to ensure long-term sensory and chemical stability of the product.

Recent studies have examined the interactions between polysaccharides and phenolic compounds in beverages derived from plant sources. Nevertheless, there is a paucity of precise reports on the influence of agavins in demonstrating these interactions. Moreover, it has been demonstrated that the interaction between polysaccharides and polyphenols in food systems is influenced by a number of factors, including pH and titratable acidity [40,41]. These interactions, which are predominantly non-covalent in nature, encompass hydrogen bonds, hydrophobic interactions, and electrostatic forces [42,43].

pH exerts a significant influence on the ionization, charge, and solubility of both components, thereby modulating their interactions [44]. It has been demonstrated that low pH conditions inhibit the aggregation and recrystallisation of starch–polyphenol mixtures [45]. Furthermore, the polysaccharide concentration has been demonstrated to exert a considerable influence on these interactions, with the most pronounced effects observed in the vicinity of the critical overlap concentration [46]. The affinity between polysaccharides and polyphenols is contingent upon their structural characteristics. Linear arabinans demonstrate a superior capacity to retain phenolic compounds in comparison to branched polysaccharides [47].

This study hypothesizes that the formation of complexes between polysaccharides and phenolic compounds can influence their bioavailability by modifying their solubility, stability, and absorption in the gastrointestinal tract. The potential for the formation of hydrogen bonds between the hydroxyl and carboxyl groups of phenolic acids, such as quinic acid and hydroxybenzoic acids, and the O-H groups of agavins is of particular interest. In some systems, this interaction with polysaccharides can protect phenolic compounds from oxidative or enzymatic degradation, which could favor their stability and prolong their availability in the body. This is particularly relevant in food matrices, where encapsulation in polysaccharide networks can improve absorption in the small intestine. Conversely, the formation of supramolecular complexes can limit the release of free phenols, reducing their solubility in aqueous media and potentially affecting their absorption in the intestine. Depending on the strength of the interaction and the capacity of the intestinal microbiome to release these compounds, this association could diminish their biological effectiveness. In summary, these interactions exert a significant influence on the physicochemical properties, quality, and nutritional value of foods by affecting the extractability, bioaccessibility, and metabolism of polyphenols in the gastrointestinal tract [48]. Therefore, it is essential to gain a thorough understanding of these interactions in order to optimize food processing techniques and elucidate the health benefits of polyphenols [49].

Furthermore, the literature on the effects of prebiotic fibers, such as inulin, on hormones related to appetite and subjective appetite has yielded equivocal results. Studies by Birkeland et al. [50] have shown that fructans and other prebiotic fibers can stimulate the secretion of intestinal hormones, such as glucagon-like peptide-1 (GLP-1) and peptide YY (PYY). These hormones play a crucial role in appetite regulation and glycemic control. Consequently, regular consumption of the aforementioned formulations can contribute to the regulation of energy metabolism and the feeling of fullness.

These associations have the potential to influence the quality, physicochemical properties, and nutritional value of foods [42]. However, to develop oak-based beverages, it is essential to have an in-depth understanding of these interactions. Therefore, it was crucial to explore possible combinations using techniques such as Fourier transform infrared spectroscopy (FT-IR).

Agavins are complex fructose polymers with various degrees of polymerization and isomers [22]. These carbohydrates display characteristic C–O, C–C, and C–H bands in the FT-IR spectra, which can be utilized for the identification and classification of agave syrups and the detection of potential adulterations [51,52]. The carbohydrate region in the FT-IR spectra offers crucial insights into the structural characteristics and interactions of agavins [52,53].

Interactions between polyphenols and macromolecules, particularly polysaccharides, play a pivotal role in the quality and nutritional value of food products [12]. These interactions are predominantly non-covalent in nature and involve the formation of hydrogen bonds and hydrophobic interactions [42]. The factors that influence these interactions include polysaccharide concentration, molecular structure, and physicochemical properties, as previously reported by Dridi and Bordenave [46] and Liu et al. [42]. To study these weak interactions, other researchers have used advanced analytical methods, such as isothermal titration calorimetry and transmission electron microscopy [54]. In this research, analytical methods such as analysis by UPLC-PDA-ESI-MS/MS and FT-IR, were combined to provide a comprehensive understanding of agavin-polyphenol complexes. These interactions have led to an increase in the capacity to inhibit angiotensin-converting enzymes. Despite the fact that there is little information in the literature on the impact of these synergistic interactions, Lee et al. [55] reported that arginine-fructose (Arg-Fru) enhances ACE inhibition, contributing to antihypertensive effects. Similarly, our study suggests that agavins, which contain fructose-based polysaccharides, may modulate ACE inhibition in oak infusions, potentially through interactions that enhance bioactive compound availability. Furthermore, these interactions have influenced the antioxidant activity of polyphenols and the sensory properties of foods, as reported by Bermudez-Oria et al. [56] and Bordenave et al. [57]. As observed in this research, the sensory acceptability of formulations from 6% agavin concentrations was significantly improved. Furthermore, the findings indicate that the incorporation of agavins appears to regulate the antioxidant response through multifaceted interactions with the phenolic compounds present in these beverages. Despite a lack of specific studies on these interactions in beverages, Medina-Larqué et al. [58] reported that the amalgamation of blueberry polyphenols and agavins in a murine model influenced the bioavailability and biological activity of these compounds. This finding suggests that agavins can affect the stability and functionality of polyphenols in different systems, which could explain the non-linear responses observed in the antioxidant activity of the analyzed formulations. However, further studies are needed to clarify the specific mechanisms of these interactions in food matrices.

In summary, these interactions can be harnessed for positive technological applications, as agavins can function as vehicles for phenolic compounds, delivering them to the large intestine, where microbial action can release and biotransform the bound phenolics. This process has the potential to influence the intestinal microbiota. However, further research is necessary to fully understand this area.

## 5. Conclusions

This study demonstrates that varying agavin concentrations significantly influence phenolic compound dynamics and antioxidant capacity. The interactions between agavins and polyphenols lead to substantial modifications in chemical profiles, as evidenced by changes in phenolic composition and antioxidant responses. These interactions exhibit a nonlinear pattern, where both quadratic and cubic effects contribute to the observed variations. The presence of agavins appears to enhance or suppress specific antioxidant mechanisms depending on the phenolic composition of each species.

Furthermore, the correlation between different antioxidant assays suggests a consistent trend, indicating that these chemical interactions uniformly affect antioxidant mechanisms. The observed variations in antioxidant capacity emphasize the role of structural modifications in phenolic compounds induced by agavin interactions. These findings provide valuable insights into how agavins can modulate bioactive properties, offering potential applications in functional food development.

Understanding these interactions expands knowledge on the synergistic or antagonistic effects between dietary fibers and polyphenols, highlighting their impact on antioxidant potential. This study underscores the importance of considering non-linear effects in the formulation of agavin-rich products to optimize their functional benefits. Future research should explore these mechanisms in biological systems to further validate their implications for health and food applications.

## Figures and Tables

**Figure 1 foods-14-00833-f001:**
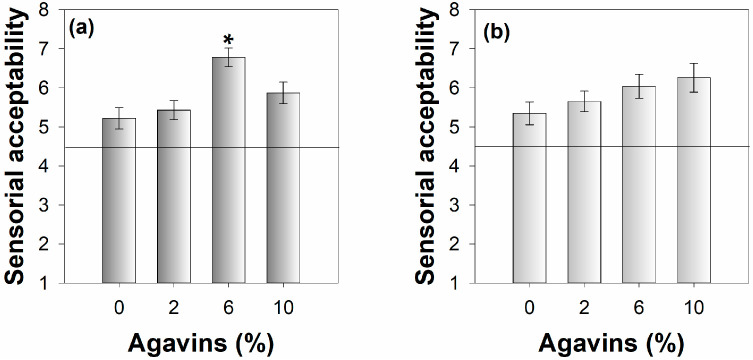
Effect of agavin concentration on sensory acceptability of oak iced tea. (**a**) *Quercus sideroxyla* and (**b**) *Quercus eduardii*. Data are expressed as mean S.E.M. * indicates statistical difference (*n* = 24, LSD, *p* < 0.05).

**Figure 2 foods-14-00833-f002:**
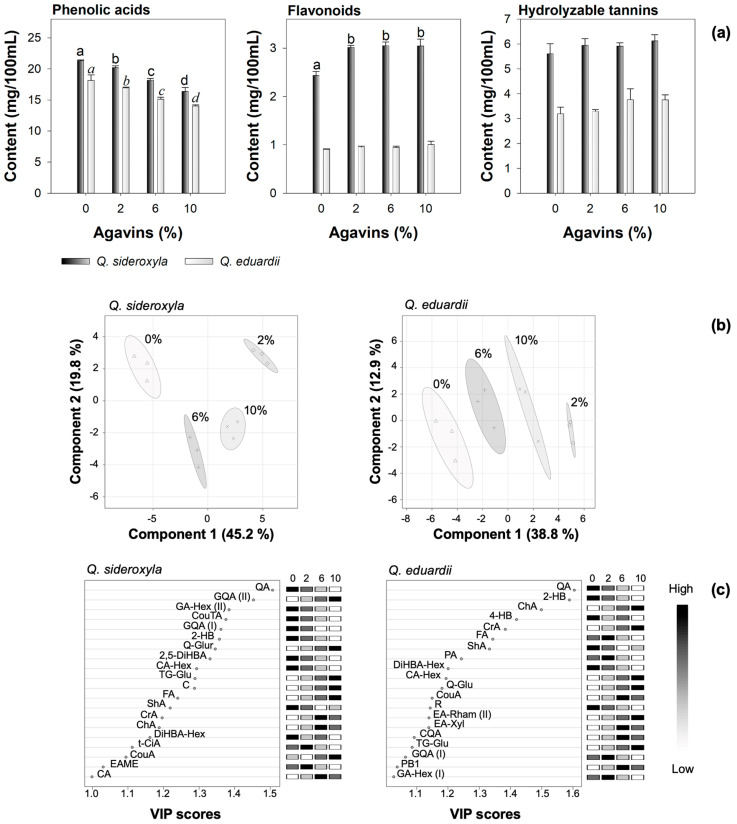
Impact of agavin concentration on phenolic profiling. (**a**) Effect on the main phenolic groups identified by UPLC-PDA-ESI-MS/MS identified by one-way ANOVA and post hoc analysis, different letters indicate significant statistical differences between the means (*n* = 3, LSD; *p* < 0.05); (**b**) partial least square discriminant analysis (PLS-DA) presented as principal component score plots; and (**c**) important features identified by PLS-DA. See Table 2 for codes of phenolic compounds present in (**c**).

**Figure 3 foods-14-00833-f003:**
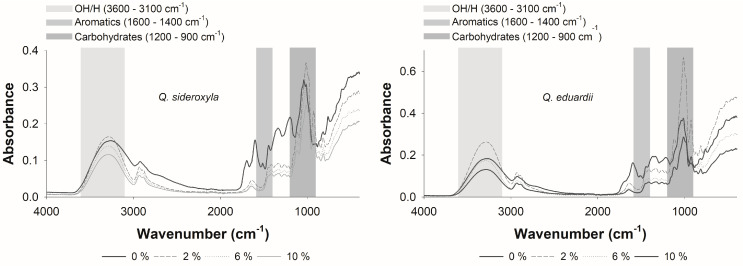
FT-IR spectrum of iced tea formulations with oak (*Quercus sideroxyla* and *Quercus eduardii*) and different concentrations of agavins.

**Figure 4 foods-14-00833-f004:**
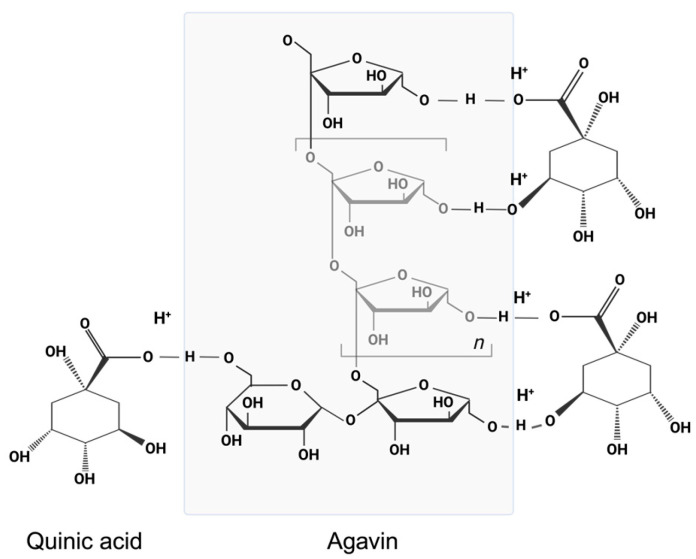
Hypothetical schematization of the chemical interaction between quinic acid and agavins represented in the illustration as a oligosaccharide model, in which a conjugated complex is formed by hydrogen bonds.

**Figure 5 foods-14-00833-f005:**
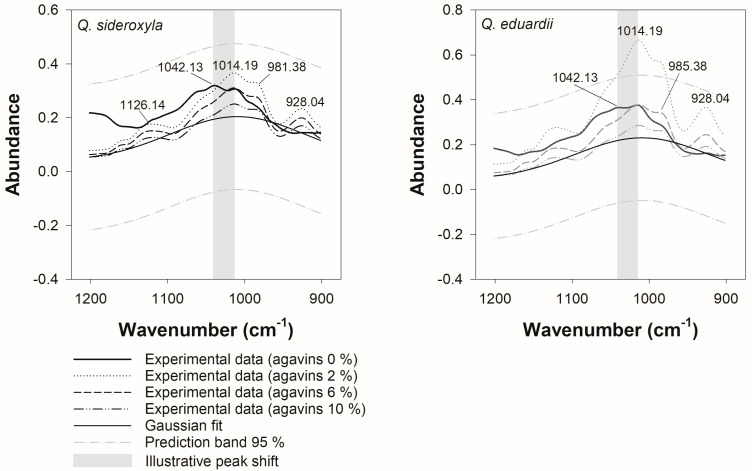
Gaussian fit of four-parameter FT-IR spectra in the 900 to 1200 cm^−^^1^ region of oak iced tea formulations with and without the addition of different concentrations of agavins.

**Table 1 foods-14-00833-t001:** Physicochemical properties of infusions sweetened with different concentrations of agavins.

Oak Species	Agavins (%)	pH	TA	°Brix
** *Quercus sideroxyla* **	0	5.42 ± 0.01 ^a^	0.01 ± 0.00 ^b^	1.13 ± 0.06 ^d^
	2	5.43 ± 0.02 ^a^	0.02 ± 0.00 ^a^	2.10 ± 0.10 ^c^
	6	5.32 ± 0.01 ^b^	0.02 ± 0.00 ^a^	5.90 ± 0.10 ^b^
	10	5.25 ± 0.01 ^c^	0.02 ± 0.00 ^a^	9.27 ± 0.12 ^a^
** *Quercus eduardii* **	0	5.84 ± 0.00 ^a^	0.01 ± 0.00 ^c^	0.95 ± 0.05 ^d^
	2	5.33 ± 0.00 ^d^	0.01 ± 0.00 ^b^	2.17 ± 0.06 ^c^
	6	5.70 ± 0.00 ^b^	0.01 ± 0.00 ^c^	6.17 ± 0.06 ^b^
	10	5.62 ± 0.00 ^c^	0.02 ± 0.00 ^a^	9.30 ± 0.10 ^a^

TA: total acidity. The results are presented as the mean ± standard deviation and different letters indicate significant statistical differences between the means (LSD, *p* < 0.05).

**Table 2 foods-14-00833-t002:** Pearson correlation matrix of the physicochemical parameters for the species of oak added with agavins.

Variable	Regardless of the Oak Species				*Quercus sideroxyla*				*Quercus eduardii*			
	Ag	pH	TTA	°Brix	Ag	pH	TA	°Brix	Ag	pH	TA	°Brix
Ag	1.00	−0.21	0.65 **	0.99 **	1.00	0.97 **	0.72 **	0.99 **	1.00	−0.51	0.78 **	0.99 **
pH		1.00	−0.49 *	−0.18		1.00	−0.57	−0.98 **		1.00	−0.14	0.01
TA			1.00	0.62 **			1.00	0.67 *			1.00	0.76 **
°Brix				1.00				1.00				1.00

Ag: agavin; TA: total acidity; * correlation is significant at the 0.05 level (2-tailed); ** correlation is significant at the 0.01 level (2-tailed).

**Table 3 foods-14-00833-t003:** Total phenolic content in infusions sweetened with different concentrations of agavins.

Oak Species	Agavins (%)	Total Phenolic Content	Total Flavonoids
** *Quercus sideroxyla* **	0	162.79 ± 6.18 ^a^	10.98 ± 0.53 ^b^
	2	134.27 ± 5.55 ^b^	13.47 ± 1.34 ^a^
	6	134.83 ± 6.92 ^b^	9.20 ± 0.31 ^c^
	10	136.79 ± 4.26 ^b^	10.09 ± 1.11 ^bc^
** *Quercus eduardii* **	0	111.76 ± 4.22 ^a^	5.28 ± 0.31 ^c^
	2	92.60 ± 3.18 ^b^	5.77 ± 0.40 ^c^
	6	105.05 ± 4.62 ^a^	7.24 ± 0.00 ^a^
	10	108.82 ± 8.96 ^a^	6.84 ± 0.13 ^b^

The results are presented as the mean ± standard deviation and different letters indicate significant statistical differences between the means (*n* = 3, LSD; *p* < 0.05).

**Table 4 foods-14-00833-t004:** Identification and quantification of extractable phenolic compound profiling in the beverages under study by UPLC-PDA-ESI-MS/MS.

Compound	*Quercus sideroxyla*				*Quercus eduardii*			
	0%	2%	6%	10%	0%	2%	6%	10%
* Phenolic acids *								
Quinic acid (QA)	12.16 ± 0.22 ^a^	10.52 ± 0.17 ^b^	8.83 ± 0.15 ^c^	6.89 ± 0.28 ^d^	9.43 ± 0.37 ^a^	8.47 ± 0.09 ^b^	7.02 ± 0.11 ^c^	5.70 ± 0.15 ^d^
Caffeoylquinic acid (CQA) *^(a)^	0.02 ± 0.00 ^b^	0.03 ± 0.00 ^a^	0.03 ± 0.00 ^a^	0.03 ± 0.00 ^a^	0.07 ± 0.01 ^a^	0.08 ± 0.00 ^a^	0.09 ± 0.01 ^a^	0.08 ± 0.01 ^a^
Chlorogenic acid (ChA)	3.68 ± 0.08 ^b^	4.32 ± 0.19 ^a^	4.54 ± 0.07 ^a^	4.44 ± 0.21 ^a^	3.03 ± 0.10 ^c^	3.27 ± 0.04 ^b^	3.47 ± 0.21 ^ab^	3.68 ± 0.08 ^a^
Cryptochlorogenic acid (CrA) *^(a)^	0.10 ± 0.00 ^b^	0.13 ± 0.01 ^a^	0.13 ± 0.01 ^a^	0.13 ± 0.00 ^a^	0.18 ± 0.01 ^b^	0.19 ± 0.01 ^ab^	0.19 ± 0.00 ^b^	0.21 ± 0.01 ^a^
Sinapoylquinic acid (SQA) *^(b)^	0.06 ± 0.00 ^b^	0.05 ± 0.01 ^b^	0.06 ± 0.01 ^ab^	0.08 ± 0.01 ^a^	0.02 ± 0.00 ^a^	0.03 ± 0.01 ^a^	0.03 ± 0.01 ^a^	0.02 ± 0.00 ^a^
Galloylquinic acid iso I (GQA (I)) *^(c)^	0.02 ± 0.00 ^a^	0.02 ± 0.00 ^a^	nd	nd	0.03 ± 0.00 ^a^	0.04 ± 0.01 ^a^	0.02 ± 0.00 ^b^	0.01 ± 0.00 ^c^
Galloylquinic acid iso II (GQA (II)) *^(c)^	0.46 ± 0.01 ^d^	0.57 ± 0.04 ^c^	0.65 ± 0.03 ^b^	0.83 ± 0.08 ^a^	0.07 ± 0.01 ^b^	0.08 ± 0.01 ^ab^	0.10 ± 0.01 ^a^	0.08 ± 0.00 ^b^
Gallic acid (GA)	1.56 ± 0.07 ^b^	2.13 ± 0.07 ^a^	2.15 ± 0.05 ^a^	1.97 ± 0.11 ^a^	2.22 ± 0.14 ^b^	2.55 ± 0.07 ^a^	2.44 ± 0.10 ^ab^	2.43 ± 0.08 ^a^
Gallic acid hexoside iso I (GA-Hex (I)) *^(c)^	0.08 ± 0.01 ^a^	0.09 ± 0.01 ^a^	0.07 ± 0.01 ^a^	0.09 ± 0.01 ^a^	0.26 ± 0.05 ^a^	0.26 ± 0.02 ^a^	0.23 ± 0.04 ^a^	0.19 ± 0.01 ^a^
Gallic acid hexoside iso II (GA-Hex (II)) *^(c)^	0.08 ± 0.00 ^a^	0.06 ± 0.01 ^b^	0.05 ± 0.00 ^c^	0.05 ± 0.01 ^bc^	0.02 ± 0.01 ^a^	0.01 ± 0.00 ^b^	0.02 ± 0.00 ^a^	0.01 ± 0.00 ^b^
2,5-Dihydroxybenzoic acid (2,5-DiHBA)	0.01 ± 0.00 ^a^	0.01 ± 0.00 ^b^	0.01 ± 0.00 ^b^	0.01 ± 0.00 ^c^	traces ^b^	0.01 ± 0.00 ^a^	traces ^a^	nd
Hydroxybenzoic acid hexoside (HBA-hex) *^(d)^	0.01 ± 0.00 ^b^	0.01 ± 0.00 ^b^	0.01 ± 0.00 ^a^	0.01 ± 0.00 ^c^	nd	nd	nd	nd
Dihydroxybenzoic acid hexoside (DiHBA-Hex) *^(d)^	0.08 ± 0.01 ^a^	0.08 ± 0.01 ^a^	0.08 ± 0.01 ^a^	0.04 ± 0.00 ^b^	0.03 ± 0.01 ^a^	0.02 ± 0.00 ^a^	0.02 ± 0.00 ^a^	0.02 ± 0.00 ^a^
4-Hydroxybenzoic acid (4-HBA)	0.01 ± 0.00 ^b^	0.02 ± 0.00 ^a^	0.01 ± 0.00 ^b^	0.01 ± 0.00 ^b^	0.01 ± 0.00 ^a^	0.01 ± 0.00 ^b^	0.01 ± 0.00 ^b^	traces ^c^
2-Hydroxybenzoic acid (2-HBA)	0.03 ± 0.00 ^a^	0.02 ± 0.00 ^a^	0.02 ± 0.00 ^ab^	0.01 ± 0.00 ^b^	0.02 ± 0.00 ^a^	0.01 ± 0.00 ^b^	0.01 ± 0.00 ^c^	0.01 ± 0.00 ^c^
Protocatechuic acid (PA)	0.13 ± 0.00 ^c^	0.16 ± 0.00 ^a^	0.15 ± 0.01 ^b^	0.12 ± 0.00 ^c^	0.35 ± 0.03 ^ab^	0.36 ± 0.00 ^a^	0.32 ± 0.01 ^b^	0.28 ± 0.02 ^c^
*p*-Coumaric acid (CouA)	0.02 ± 0.00 ^a^	0.02 ± 0.00 ^a^	0.02 ± 0.00 ^a^	0.02 ± 0.00 ^a^	0.10 ± 0.00 ^b^	0.10 ± 0.00 ^b^	0.10 ± 0.00 ^a^	0.10 ± 0.00 ^a^
Coumaric tartaric acid (CouTA) *^(e)^	0.01 ± 0.00 ^a^	traces ^b^	traces ^c^	traces ^c^	nd	nd	nd	nd
Caffeic acid hexoside (CA-hex) *^(f)^	0.01 ± 0.00 ^a^	0.01 ± 0.00 ^b^	0.01 ± 0.00 ^b^	0.01 ± 0.00 ^c^	traces ^a^	traces ^a^	traces ^a^	traces ^a^
Caffeic acid (CA)	0.01 ± 0.00 ^b^	0.02 ± 0.00 ^a^	0.02 ± 0.00 ^a^	0.02 ± 0.00 ^a^	nd	nd	nd	nd
Ferulic acid (FA)	traces ^a^	traces ^a^	traces ^a^	traces ^b^	traces ^a^	traces ^a^	traces ^a^	traces ^b^
*trans*-cinnamic acid (t-CiA)	0.10 ± 0.00 ^a^	0.10 ± 0.00 ^ab^	0.10 ± 0.00 ^b^	0.10 ± 0.00 ^b^	0.10 ± 0.00 ^b^	0.10 ± 0.00 ^a^	0.10 ± 0.00 ^a^	0.10 ± 0.00 ^a^
Shikimic acid (ShA)	2.76 ± 0.34 ^a^	1.91 ± 0.02 ^b^	1.33 ± 0.10 ^d^	1.62 ± 0.11 ^c^	2.41 ± 0.24 ^a^	1.55 ± 0.19 ^b^	1.13 ± 0.14 ^c^	1.29 ± 0.06 ^c^
* Flavonoids *								
Procyanidin (PB1)	0.06 ± 0.00 ^b^	0.08 ± 0.01 ^a^	0.07 ± 0.00 ^a^	0.06 ± 0.00 ^b^	0.01 ± 0.00 ^b^	0.01 ± 0.00 ^b^	0.02 ± 0.01 ^a^	0.02 ± 0.00 ^a^
Catechin (C)	0.71 ± 0.02 ^c^	0.81 ± 0.00 ^b^	0.82 ± 0.02 ^ab^	0.84 ± 0.02 ^a^	0.22 ± 0.02 ^a^	0.22 ± 0.00 ^a^	0.21 ± 0.01 ^a^	0.23 ± 0.02 ^a^
(epi)-catechin (epi-C)	0.08 ± 0.01 ^a^	0.09 ± 0.01 ^a^	0.08 ± 0.00 ^a^	0.09 ± 0.02 ^a^	0.05 ± 0.01 ^a^	0.06 ± 0.01 ^a^	0.06 ± 0.00 ^a^	0.05 ± 0.01 ^a^
Myricetin hexoside (M-Hex) *^(g)^	nd	nd	nd	nd	traces ^b^	traces ^b^	traces ^c^	traces ^a^
Rutin (R)	0.16 ± 0.01 ^b^	0.19 ± 0.01 ^a^	0.20 ± 0.02 ^a^	0.18 ± 0.00 ^a^	0.12 ± 0.01 ^a^	0.11 ± 0.01 ^a^	0.10 ± 0.00 ^a^	0.10 ± 0.00 ^a^
Quercetin 3-O-β-glucuronide (Q-glur)	0.75 ± 0.03 ^c^	0.90 ± 0.02 ^b^	0.92 ± 0.01 ^b^	0.96 ± 0.01 ^a^	0.25 ± 0.01 ^a^	0.28 ± 0.01 ^a^	0.29 ± 0.01 ^a^	0.28 ± 0.03 a
Quercetin 3-O-glucoside (Q-glu)	0.62 ± 0.01 ^b^	0.89 ± 0.01 ^a^	0.89 ± 0.08 ^a^	0.84 ± 0.11 ^a^	0.17 ± 0.02 ^a^	0.20 ± 0.01 ^a^	0.20 ± 0.02 ^a^	0.23 ± 0.03 ^a^
Quercetin rhamnoside (QR) ^*(h)^	nd	nd	nd	nd	0.06 ± 0.01 ^b^	0.06 ± 0.01 ^b^	0.04 ± 0.00 ^c^	0.08 ± 0.01 ^a^
Quercetin (Q)	0.10 ± 0.00 ^a^	0.10 ± 0.00 ^a^	0.10 ± 0.00 ^a^	0.10 ± 0.00 ^a^	nd	nd	nd	nd
Kaempferol 3-O-glucoside (K-glu) *^(i)^	0.02 ± 0.00 ^a^	0.02 ± 0.00 ^a^	0.02 ± 0.00 ^a^	0.02 ± 0.00 ^a^	0.10 ± 0.00 ^a^	0.10 ± 0.00 ^a^	0.10 ± 0.00 ^a^	0.10 ± 0.00 ^a^
Kaempferol rutinoside (KR) *^(i)^	0.03 ± 0.00 ^a^	0.02 ± 0.00 ^b^	0.03 ± 0.01 ^ab^	0.03 ± 0.00 ^a^	traces ^a^	traces ^a^	traces ^a^	traces ^b^
Phloridzin dihydrate (Ph)	0.10 ± 0.00 ^a^	0.10 ± 0.00 ^a^	0.10 ± 0.00 ^a^	0.10 ± 0.00 ^a^	0.10 ± 0.00 ^b^	0.10 ± 0.00 ^a^	0.10 ± 0.00 ^b^	0.10 ± 0.00 b
* Hydrolyzable tannins *								
Trigalloyl glucoside (TG-glu) *^(c)^	0.78 ± 0.03 ^c^	0.84 ± 0.05 ^bc^	0.85 ± 0.03 ^b^	0.93 ± 0.02 ^a^	0.49 ± 0.07 ^a^	0.51 ± 0.05 ^a^	0.57 ± 0.06 ^a^	0.61 ± 0.07 ^a^
Ellagic acid xyloside (EA-Xyl) *^(j)^	0.43 ± 0.06 ^a^	0.48 ± 0.04 ^a^	0.46 ± 0.05 ^a^	0.45 ± 0.02 ^a^	1.09 ± 0.11 ^a^	1.14 ± 0.08 ^a^	1.34 ± 0.12 ^a^	1.32 ± 0.15 ^a^
Ellagic acid rhamnoside iso I (EA-Rham (I)) *^(j)^	0.18 ± 0.01 ^a^	0.16 ± 0.01 ^a^	0.16 ± 0.02 ^a^	0.17 ± 0.02 ^a^	0.48 ± 0.05 ^ab^	0.44 ± 0.02 ^b^	0.54 ± 0.03 ^a^	0.48 ± 0.07 ^ab^
Ellagic acid rhamnoside iso II (EA-Rham (II)) *^(j)^	0.12 ± 0.02 ^a^	0.13 ± 0.01 ^a^	0.14 ± 0.02 ^a^	0.14 ± 0.01 ^a^	0.17 ± 0.02 ^a^	0.18 ± 0.02 ^a^	0.20 ± 0.02 ^a^	0.21 ± 0.02 ^a^
Ellagic acid glucoside (EA-glu) *^(j)^	4.09 ± 0.36 ^a^	4.32 ± 0.17 ^a^	4.27 ± 0.15 ^a^	4.43 ± 0.20 ^a^	0.94 ± 0.10 ^a^	0.99 ± 0.07 ^a^	1.08 ± 0.22 ^a^	1.11 ± 0.07 ^a^
Ellagic acid derivative (EA-der) *^(j)^	0.10 ± 0.00 ^b^	0.10 ± 0.00 ^a^	0.10 ± 0.00 ^a^	0.10 ± 0.00 ^a^	traces ^b^	0.10 ± 0.00 ^a^	0.10 ± 0.00 ^a^	traces ^b^
Ellagic acid methyl ether (EAME) *^(j)^	0.10 ± 0.00 ^a^	0.10 ± 0.00 ^a^	traces ^b^	traces ^b^	0.10 ± 0.00 ^b^	0.02 ± 0.00 ^a^	0.02 ± 0.00 ^a^	0.10 ± 0.00 ^b^

RT denotes retention time; results are expressed as average ± standard deviation (*n* = 3, Tukey; *p* < 0.05); * compounds identified on the basis of their major transitions and quantified on the base of qualifier ion and the literals denote as equivalent to (a) chlorogenic acid, (b) sinapic acid, (c) gallic acid, (d) 4-hydroxybenzoic acid, (e) coumaric acid, (f) caffeic acid, (g) myricetin, (h) quercetin, (i) kaempferol, and (j) ellagic acid; nd denotes not detected.

**Table 5 foods-14-00833-t005:** Nutraceutical potential of oak infusions sweetened with different concentrations of agavins.

Oak Species	Agavins (%)	ACE Inhibition (%)	ABTS	FRAP	ORAC
*Quercus sideroxyla*	0	77.93 ± 0.22 ^b^	220.01 ± 3.20 ^a^	143.22 ± 7.94 ^c^	133.94 ± 8.66 ^c^
	2	94.47 ± 0.36 ^a^	195.50 ± 1.76 ^b^	389.30 ± 20.97 ^a^	215.12 ± 1.08 ^b^
	6	94.82 ± 0.59 ^a^	217.39 ± 2.32 ^a^	202.64 ± 7.27 ^b^	373.68 ± 34.09 ^a^
	10	94.41 ± 0.46 ^a^	220.48 ± 0.00 ^a^	184.47 ± 15.17 ^b^	158.83 ± 17.32 ^c^
*Quercus eduardii*	0	80.11 ± 0.45 ^b^	220.37 ± 1.09 ^b^	173.98 ± 2.42 ^a^	204.83 ± 4.87 ^a^
	2	95.17 ± 0.35 ^a^	224.53 ± 1.09 ^a^	120.15 ± 5.28 ^c^	159.19 ± 13.71 ^b^
	6	95.48 ± 0.24 ^a^	220.72 ± 0.74 ^b^	161.40 ± 9.91 ^b^	202.67 ± 17.68 ^a^
	10	95.64 ± 0.27 ^a^	220.48 ± 1.43 ^b^	155.80 ± 12.64 ^b^	148.01 ± 10.82 ^b^

Results are presented as mean ± standard deviation and different letters indicate significant statistical differences between means (LSD, *p* < 0.05).

**Table 6 foods-14-00833-t006:** Fit cubic models for antioxidant activity as a function of agavin concentration and their interactions with polyphenols.

Formulation	Equation	Goodness of Fit
*Q. sideroxyla*	**ABTS** = 220.00 − 25.22 X + 6.54 X^2^ − 0.4 X^3^	0.98
	**FRAP** = 143.22 + 219.88 X − 55.12 X^2^ + 3.53 X^3^	0.98
	**ORAC** = 133.93 + 27.09 X + 9.10 X^2^ − 1.15 X^3^	0.97
*Q. eduardii*	**ABTS** = 220.36 + 3.84 X−1.00 X^2^ + 0.06 X^3^	0.95
	**FRAP** = 173.97 − 48.52 X + 12.34 X^2^ − 0.77 X^3^	0.97
	**ORAC** = 204.83 − 44.46 X + 12.56 X^2^ − 0.87 X^3^	0.98

## Data Availability

The original contributions presented in this study are included in the article. Further inquiries can be directed at the corresponding author.

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
