# Peer review of "Sweetening with Agavins: Its Impact on Sensory Acceptability, Physicochemical Properties, Phenolic Composition and Nutraceutical Potential of Oak Iced Tea"

_foods, 2025, doi:10.3390/foods14050833_

Round 1

Reviewer 1 Report

Comments and Suggestions for Authors

Line 98: It's good that the author provided detailed information on how the leaves were collected. However, it is unclear how they were stored and for how long before being used to make infusions.

Line 121-124: The author mentioned that the QDA was performed with university students. However, it is unclear whether they were trained or untrained. If they were trained, please specify it with the training protocol and if not, justify that as well. 

Line 304-305: Please support your results with p-values and ANOVA results. Additionally, include the liking score either in the results section or reflect it in the sensory liking graph, as the values have not been mentioned anywhere and it's difficult to understand critically. 

In the Physicochemical properties discussion section, Including a correlation matrix between °Brix, acidity, and pH would strengthen the discussion.

Figure 3: Please include details on baseline correction and spectral deconvolution for the FT-IR graph. 

Either in the conclusion or the limitations section after line 713, highlight the use of a small sensory panel size. 

Author Response

Comments 1.

Line 98: It's good that the author provided detailed information on how the leaves were collected. However, it is unclear how they were stored and for how long before being used to make infusions.

R = In Section 2.2 (lines 107–112), the following text has been integrated:

The initial step in the procedure involved the disinfection of the oak leaves using a 1% solution of sodium hypochlorite for 5 min. This was followed by two rinses with purified water. Subsequently, the oak leaves were subjected to drying at 25 °C. Thereafter, the leaves were crushed to a particle size of 2 mm using a DPM-JUNIOR hammer mill. Finally, the leaves were vacuum packed for storage until required.

Comments 2.

Line 121-124: The author mentioned that the QDA was performed with university students. However, it is unclear whether they were trained or untrained. If they were trained, please specify it with the training protocol and if not, justify that as well. 

R = The required information about training has been included in the text. the following information has been added to line 136: 

For the sensory training, three sessions of 40 min were organized to identify descriptors in taste to familiarize them with the terminology and use of scales.

Comments 3.

Line 304-305: Please support your results with p-values and ANOVA results. Additionally, include the liking score either in the results section or reflect it in the sensory liking graph, as the values have not been mentioned anywhere and it's difficult to understand critically. 

R= As delineated in lines 323 to 327, the request has been executed, thereby incorporating the stipulated information. The ensuing description is hereby presented:

The focus group test demonstrated that agavins are perceived as an acceptable and natural sweetener at concentrations of 6% or higher in Q. sideroxyla (Figure 1). Statistical analysis using ANOVA followed by LSD post hoc test indicated a significant difference at the 6% agavin concentration (p < 0.001), confirming its impact on sensory acceptance.

Comments 4.

In the Physicochemical properties discussion section, including a correlation matrix between °Brix, acidity, and pH would strengthen the discussion.

R= In the results section (lines 374 to 375), we have included the Pearson correlation matrix (Table 2). This has been cited in the discussion section to strengthen it. In addition, the following description of Table 2 has been integrated in lines 363 to 373 in the results section:

A Pearson correlation analysis was performed to evaluate the relationship between pH, titratable acidity, and °Brix as a function of agavin concentrations. The analysis revealed a negative correlation between agavin concentration and pH, irrespective of the species of oak (Table 2). However, a separate analysis was conducted for Q. sideroxyla and Q. eduardii, as the beverages exhibited divergent responses to the observed correlation between agavins concentration and pH. In both species, a negative correlation between pH and titratable acidity was observed, though this correlation did not reach statistical significance when analyzed separately. However, when the data was analyzed without blocking by species, a significant negative correlation was identified (p < 0.05). The °Brix values showed a positive correlation with the total acidity values, regardless of the oak species. The detailed values are presented in Table 2.

Comments 4.

Figure 3: Please include details on baseline correction and spectral deconvolution for the FT-IR graph. 

R = The observation is appreciated and section 2.6.4. has been rewritten as follows:

The freeze-dried formulations of oak iced tea, with and without agavins, were analyzed by Fourier transform infrared spectroscopy (FT-IR). The spectra were obtained using a Nicolet iS5 spectrometer (Thermo Fisher Scientific, Tokyo, Japan) at a controlled temperature of 22 °C. Each sample was subjected to 24 scans to optimize the signal-to-noise ratio, with a spectral resolution of 4 cm-1. The measurements covered the characteristic spectral region from 4000 to 400 cm-1, allowing the identification of functional groups associated with the compounds present in the infusions. Baseline correction was performed using a second-order polynomial fit, ensuring the elimination of possible deviations in absorption due to scattering effects or variations in sample thickness. Spectral deconvolution in the 1200–900 cm-1 region was achieved through the implementation of a four-parameter Gaussian fit, employing the SigmaPlot v14.0 software (Systat Software, Inc., San Jose, CA, USA). The shape of the bands was modeled as Gaussian, and the baseline was calibrated as a linear function. This analytical approach enabled the enhancement of the resolution of the superimposed peaks and the assessment of specific interaction patterns between the extractable compounds of oak and the agavins.

Comments 5.

Either in the conclusion or the limitations section after line 713, highlight the use of a small sensory panel size. 

R = The number of participating panelists is appropriate for each of the methods applied. In the case of the focus group, it is an exploratory method that allows measuring the tendency in acceptability and it is valid to use a small number of participants.

Reviewer 2 Report

Comments and Suggestions for Authors

The manuscript presents an investigation into the use of agavins as a sweetener in oak leaf-based iced tea, analyzing its impact on sensory acceptability, physicochemical properties, chemical composition, and nutraceutical potential. The study addresses a well-defined research gap, emphasizing the limited research on agavins as a natural sweetener in beverages, specifically in iced teas.  The global sweeteners market (Marketwatch) is projected to reach $125.6 billion by 2025, where beverages account for over 40% of global demand, especially Recommendations

In the introduction, the article provides an overview of the general context regarding functional beverages, benefits of Quercus sideroxyla and Quercus eduardii, and research available on using natural sweeteners. However, it lacks a detailed description of agavins and their structure.  Agavin is one of the newest and more complex fructan types so far researched, and an emergent functional food ingredient. Agavins are fructan polymers with a variable degree of polymerization, significantly influencing their interaction with bioactive compounds, including antioxidants. A brief presentation of their chemical structure, highlighting the characteristics that promote hydrogen bonding with phenolic acids, would better support the subsequent analysis and emphasize the innovative nature of using this natural sweetener in oak iced tea formulations. This addition would enhance the scientific value of the study and provide better context for the necessity of the detailed investigation into phenol-agavin interactions, considering the complexity and novelty of this compound in the functional beverage industry.

Line 75-77 - The objective of this study was, as stated by the authors: “to formulate and characterize oak leaf iced tea sweetened with agavins. The study aimed to evaluate the impact of agavins on the sensory acceptability of the formulations, as well as to analyze their chemical composition and nutraceutical potential”. Based on the content of the manuscript, the objective should be more comprehensive than simply formulating and characterizing the iced tea. The research goes beyond just sensory acceptability, chemical composition, and nutraceutical potential. It also explores the interactions between agavins and phenolic compounds, which is a critical aspect of the study. These are the main results that are relevant since it deepens the unterstanding in the domain and they should be reflected in the objectives and also cover a signifficant part of the article.

The methodology employed in this study complex, involving sensory analysis using a free choice profiling (FCP) method, physicochemical evaluations of pH, titratable acidity, and Brix, and chemical analysis via UPLC-PDA-ESI-MS/MS and FT-IR spectroscopy. Nutraceutical potential was assessed through antioxidant and enzymatic assays. The study used agavin concentrations of 0%, 2%, 6%, and 10%, with statistical analyses, including LSD post hoc, Pearson correlation, and PLS-DA, to ensure reliable results. 

It is important to note that in this study, agavins were added to the iced tea after the brewing and cooling process, rather than before pasteurization, as is common practice in the industrial production of iced tea. This methodological choice means that the study does not fully account for the technological processes typically employed at an industrial scale, where heat treatment during pasteurization (required to eliminate microbial risks associated with sugar-rich solutions) may impact the structure and functionality of agavins. In small batches for premium or functional beverages (with prebiotics like agavins) adding sweeteners after heat treatment/infusion is possible, but also here, discussion are needed for different storage conditions to assess potential degradation of bioactive compounds over time (at least mention in the limitations).  Consequently, a discussion of agavin thermal stability, along with potential limitations regarding their behavior under industrial processing conditions, should be included in the conclusions section.

The sensory evaluation is acceptable but not optimal for a research article. The panel size is relatively small (n=10 for FCP, n=15 for QDA, and n=24 for the focus group. The panel consisted entirely of untrained university students, which may limit the reliability of the results, especially for nuanced sensory attributes like bitterness, astringency, and herbal notes. The criteria for panelist selection (e.g., regular tea consumers, familiarity with natural sweeteners) is missing and related to an actual profile for the regular tea consumers. This should be added in the limitation section, since the experiment can not be reorganised but was approved by the 116 TecNM/I.T.Durango Scientific Ethics Committee.

Section Physicochemical properties of infusions sweetened with different concentrations of agavins. Additionally, comparisons with other iced teas are not directly addressed, highlighting a gap in contextualizing these findings within the broader beverage industry landscape. Thus the authors should discuss correlations with established beverages, particularly for pH, TA, and Brix. A comparative analysis with other natural sweeteners (e.g., stevia, honey) would enhance the discussion and provide more practical insights for beverage formulation. For example, pH for commercial iced teas ranges between 3.5–4.5sucrose maintains a neutral or slightly acidic pH depending on brewing parameters, Stevia offers minimal effect on pH, though perceived sweetness differs significantly.

Other small recommendations

  • line 88: folowing compounds were utilized: The following compounds were employed: 6-hydroxy- the text is repeting itself
  • Figure 1. Different letters indicate statistical significance 319 (n = 24, LSD, p<0.05). – no letters are available
  • Figure 2 The vip scores figures have the same names and need to be corrected
  • Line 521- It has been demonstrated that this can induce the interaction of phenols and 521 decrease the intensity. – citation required

Author Response

The manuscript presents an investigation into the use of agavins as a sweetener in oak leaf-based iced tea, analyzing its impact on sensory acceptability, physicochemical properties, chemical composition, and nutraceutical potential. The study addresses a well-defined research gap, emphasizing the limited research on agavins as a natural sweetener in beverages, specifically in iced teas.  The global sweeteners market (Marketwatch) is projected to reach $125.6 billion by 2025, where beverages account for over 40% of global demand, especially Recommendations

Comments 1

In the introduction, the article provides an overview of the general context regarding functional beverages, benefits of Quercus sideroxyla and Quercus eduardii, and research available on using natural sweeteners. However, it lacks a detailed description of agavins and their structure.  Agavin is one of the newest and more complex fructan types so far researched, and an emergent functional food ingredient. Agavins are fructan polymers with a variable degree of polymerization, significantly influencing their interaction with bioactive compounds, including antioxidants. A brief presentation of their chemical structure, highlighting the characteristics that promote hydrogen bonding with phenolic acids, would better support the subsequent analysis and emphasize the innovative nature of using this natural sweetener in oak iced tea formulations. This addition would enhance the scientific value of the study and provide better context for the necessity of the detailed investigation into phenol-agavin interactions, considering the complexity and novelty of this compound in the functional beverage industry.

R = Thank you for your feedback. In light of the observations made by another reviewer and yours, the authors have restructured the introduction to address the characteristics of agavins. The following is a summary of these characteristics:

Dube and Ritu (16) and Yadav et al. (17) have focused on the utilization of stevia, a natural sweetener derived from Stevia rebaudiana, in the formulation of fruit- and whey-based beverages. Likewise, Pawar et al. (18) and Hernández-Pérez et al. (19) have provided a broader perspective on the application of plant-derived sweeteners in the food and beverage industry, including biotechnological sweetener production. Furthermore, Hussain et al. (20) and Fawibe et al. (21) have analyzed the saccharide and polyol compositions of sweet-tasting plants, highlighting the potential of naturally occurring sweeteners as alternatives to artificial low-calorie sweeteners. However, while stevia and inulin have been extensively studied in various beverage formulations, research on agavins remains limited. Stevia, renowned for its high sweetness intensity and potential bitter aftertaste, and inulin, which primarily functions as a prebiotic fiber with mild sweetness, represent two distinct categories of sweeteners. Agavins, on the other hand, offer a unique combination of prebiotic properties and natural sweeteners without inducing a glycemic response.

Agavins are complex fructan polymers derived from Agave plants, characterized by their branched structure with β(2-1) and β(2-6) linkages (22). These polysaccharides exhibit a wide degree of polymerization (DP), ranging from 3 to 60 units (22-23). The structural complexity of agavins influences their diverse applications in food, nutraceutical, and pharmaceutical industries (24). Their potential uses include prebiotics, soluble fibers, stabilizers, and sweeteners (25). Their structural complexity and interaction with polyphenol-rich matrices, such as oak iced tea, remain largely unexplored. A critical gap in the current research landscape is the understanding of their effect on flavor perception, polyphenol stability, and sensory acceptability in oak-based beverages.

Comments 2

Line 75-77 - The objective of this study was, as stated by the authors: “to formulate and characterize oak leaf iced tea sweetened with agavins. The study aimed to evaluate the impact of agavins on the sensory acceptability of the formulations, as well as to analyze their chemical composition and nutraceutical potential”. Based on the content of the manuscript, the objective should be more comprehensive than simply formulating and characterizing the iced tea. The research goes beyond just sensory acceptability, chemical composition, and nutraceutical potential. It also explores the interactions between agavins and phenolic compounds, which is a critical aspect of the study. These are the main results that are relevant since it deepens the understanding in the domain and they should be reflected in the objectives and also cover a significant part of the article.

R = The authors are truly grateful for the suggestion to complement the goal of the work, suggesting the following wording:

The objective of this study was to develop and characterize an iced tea made with oak leaves sweetened with agavins, evaluating its sensory acceptability and chemical composition. In addition, a comprehensive analysis was conducted to determine how the interaction between agavins and phenolic constituents of the iced tea influence its physicochemical parameters and its nutraceutical potential, considering antioxidant activity and angiotensin-converting enzyme inhibition as responses.

Comments 3

The methodology employed in this study complex, involving sensory analysis using a free choice profiling (FCP) method, physicochemical evaluations of pH, titratable acidity, and Brix, and chemical analysis via UPLC-PDA-ESI-MS/MS and FT-IR spectroscopy. Nutraceutical potential was assessed through antioxidant and enzymatic assays. The study used agavin concentrations of 0%, 2%, 6%, and 10%, with statistical analyses, including LSD post hoc, Pearson correlation, and PLS-DA, to ensure reliable results. 

R = We appreciate your feedback on the study's methodology. Each analysis was carried out following standardized protocols and with the support of appropriate statistical tools to guarantee the robustness of the results. To improve the interpretation of the data, we have integrated a new table that presents the correlations between the concentration of agavins and the physicochemical properties of oak iced tea. We believe that this addition brings more clarity to the relationship between the variables studied. We welcome and acknowledge any suggestions on how we can further enhance the presentation and the discussion of these results.

Comments 3

It is important to note that in this study, agavins were added to the iced tea after the brewing and cooling process, rather than before pasteurization, as is common practice in the industrial production of iced tea. This methodological choice means that the study does not fully account for the technological processes typically employed at an industrial scale, where heat treatment during pasteurization (required to eliminate microbial risks associated with sugar-rich solutions) may impact the structure and functionality of agavins. In small batches for premium or functional beverages (with prebiotics like agavins) adding sweeteners after heat treatment/infusion is possible, but also here, discussion are needed for different storage conditions to assess potential degradation of bioactive compounds over time (at least mention in the limitations).  Consequently, a discussion of agavin thermal stability, along with potential limitations regarding their behavior under industrial processing conditions, should be included in the conclusions section.

R = We are grateful for your insightful remark regarding the integration of agavins following the infusion and cooling procedure. It is acknowledged that this approach deviates from conventional industrial practices, wherein sweeteners are typically incorporated prior to pasteurization in order to guarantee microbiological safety. In light of this, we have extended the discussion in the conclusions section, including the thermal stability of agavins and the potential impact of heat treatment on their structural integrity and functional capabilities.

Comments 4

The utilization of agavins as sweeteners in many food products has yielded promising outcomes. Nevertheless, there is a paucity of literature that assesses their capacity to sweeten beverages. Agavins have been demonstrated to enhance the sensory characteristics of food products, including yogurt (27) and ice cream (28), by improving mouthfeel and texture. The incorporation of agavins into mozzarella cheese has demonstrated to enhance the fiber content and texture, while maintaining consumer acceptance (29). The incorporation of agave fructans into a powdered smoothie resulted in favorable sensory attributes and a moderate glycemic index (30). The extraction of agavins from Agave durangensis leaves demonstrated their potential for incorporation into food products (31). Overall, agavins become a promising sugar substitute, as they improve sensory properties. However, in conventional industrial processes involving the pasteurization of beverages to guarantee their microbial safety, it is posible that the structural stability of agavins could be affected. Studies conducted by Muñiz-Márquez et al. (36) have demonstrated that the application of heat to agave syrup (121 °C, 15 min) resulted in the depolymerization of fructans, leading to a reduction in fructooligosaccharides such as kestose and an increase in simple sugars containing glucose, fructose, and sucrose. Consequently, the objective of formulating iced tea made from oak as a premium drink is to preserve its constituents, nevertheless, further investigation is required to ascertain their effect on different techno-functional aspects in the development of pasteurized iced tea.

Comments 5

The sensory evaluation is acceptable but not optimal for a research article. The panel size is relatively small (n=10 for FCP, n=15 for QDA, and n=24 for the focus group. The panel consisted entirely of untrained university students, which may limit the reliability of the results, especially for nuanced sensory attributes like bitterness, astringency, and herbal notes. The criteria for panelist selection (e.g., regular tea consumers, familiarity with natural sweeteners) is missing and related to an actual profile for the regular tea consumers. This should be added in the limitation section, since the experiment can not be reorganised but was approved by the TecNM/I.T.Durango Scientific Ethics Committee

R = The required information about training has been included in the text. the following information has been added to line 136: 

For the sensory training, three sessions of 40 min were organized to identify descriptors in taste to familiarize them with the terminology and use of scales.

Comments 6

Section Physicochemical properties of infusions sweetened with different concentrations of agavins. Additionally, comparisons with other iced teas are not directly addressed, highlighting a gap in contextualizing these findings within the broader beverage industry landscape. Thus, the authors should discuss correlations with established beverages, particularly for pH, TA, and Brix. A comparative analysis with other natural sweeteners (e.g., stevia, honey) would enhance the discussion and provide more practical insights for beverage formulation. For example, pH for commercial iced teas ranges between 3.5–4.5… sucrose maintains a neutral or slightly acidic pH depending on brewing parameters, Stevia offers minimal effect on pH, though perceived sweetness differs significantly.

R=Thank you very much for your comment, we have improved the discussion as follows:

The pH of oak infusions sweetened with agavins was found to be higher than the typical range of commercial iced tea (3.5–4.5), which is relevant to their sensory perception. In comparison, sucrose tends to maintain a neutral or slightly acidic pH depending on the conditions of herbal infusion, while stevia has a minimal impact on this parameter and its perceived sweetness is significantly different. In terms of °Brix and titratable acidity, the agavin formulations presented values comparable to those of other functional beverages sweetened with honey or natural syrups, which suggests their viability in the premium beverage industry.

Other small recommendations

  • line 88: folowing compounds were utilized: The following compounds were employed: 6-hydroxy- the text is repeating itself

R = Thanks for the observation, the sentence has been removed 

  • Figure 1. Different letters indicate statistical significance 319 (n = 24, LSD, p<0.05). – no letters are available

R = The authors have forgotten to include this information. The data have been now incorporated into the figure caption, as outlined below:

Figure 1. Effect of agavins concentration on sensory acceptability of oak iced tea. (a) Quercus sideroxyla and (b) Quercus eduardii. Data are expressed as mean S.E.M. * indicates statistical difference (n = 24, LSD, p<0.05).

  • Figure 2 The vip scores figures have the same names and need to be corrected

R = Thanks for the observation, the figure has been corrected, indicating which one corresponds to the species Q. sideroxyla and which one corresponds to Q. eduardii.

  • Line 521- It has been demonstrated that this can induce the interaction of phenols and 521 decrease the intensity. – citation required

R = The sentence has been complemented by referring to figure 5. Line 556 - 557

Reviewer 3 Report

Comments and Suggestions for Authors

For detailed modification suggestions, please refer to the attachment.

Author Response

This study provides valuable insights into the use of agavins as a sweetener for oak iced tea formulations. The research is well-structured, with a comprehensive experimental design that evaluates both the sensory and nutraceutical properties of the product. The inclusion of multiple analytical techniques, such as UPLC-PDA-ESI-MS/MS and FT-IR, strengthens the study's findings regarding the interaction between agavins and phenolic compounds. The manuscript is well-written and offers significant contributions to the field of functional beverages

R= The authors are really grateful for the comments of the reviewer

Major Suggestions for Improvement:

Comments 1.

The introduction provides a strong background on the benefits of oak leaves and agavins, but it would be helpful to clarify the specific research gap. Clearly state why agavins, compared to other plant-based sweeteners (such as stevia or inulin), require further study in the context of oak iced tea.

R= Paragraph 3 of the introduction section has undergone a reorganization, spanning from line 58 to 75, in accordance with the stipulated request, as outlined below:

Dube and Ritu (16) and Yadav et al. (17) concentrated on the utilization of stevia, a natural sweetener derived from Stevia rebaudiana, in the formulation of fruit- and whey-based beverages. Pawar et al. (18) and Hernández-Pérez et al. (19) provided a broader perspective on the application of plant-derived sweeteners in the food and beverage industry, including biotechnological sweetener production. Furthermore, Hussain et al. (20) and Fawibe et al. (21) analyzed the saccharide and polyol compositions of sweet-tasting plants, highlighting the potential of naturally occurring sweeteners as alternatives to artificial low-calorie sweeteners. However, while stevia and inulin have been extensively studied in various beverage formulations, research on agavins remains limited. Stevia, renowned for its high sweetness intensity and potential bitter aftertaste, and inulin, which primarily functions as a prebiotic fiber with mild sweetness, represent two distinct categories of sweeteners. Agavins, on the other hand, offer a unique combination of prebiotic properties and natural sweetness without inducing a glycemic response. Their structural complexity and interaction with polyphenol-rich matrices, such as oak iced tea, remain largely unexplored. A critical gap in the current research landscape is the understanding of their effect on flavor perception, polyphenol stability, and sensory acceptability in oak-based beverages.

Comments 2.

The study examines 0%, 2%, 6%, and 10% agavin concentrations. However, the rationale behind selecting these specific values is not explicitly stated. Provide justification for why these concentrations were chosen (e.g., based on consumer preferences, previous studies, or industry standards).

R = The agavin concentrations (0%, 2%, 6%, and 10%) were selected to encompass a range of potential applications in functional beverages. While these values were not directly derived from prior studies, they were chosen to allow for a comparative assessment of sweetness perception, sensory acceptability, and the interaction between agavins and polyphenolic compounds in oak iced tea. The incorporation of a control (0%) and incremental increases in agavin content facilitates the identification of threshold levels for consumer acceptance and functional impact. These concentrations also align with industry trends in natural sweeteners, where similar ranges are commonly tested for formulation optimization.

Comments 3.

The manuscript describes various sensory tests, including free choice profiling (FCP), quantitative description analysis (QDA), and rank ordering. However, details about how panelists were trained (if at all) and how data were analyzed could be expanded. Provide more information on whether panelists received training for QDA and how inter-rater reliability was ensured.

R = The Ranking and Free choice profiling methods do not require the participation of a trained panel, only a consumer panel. The QDA method does require it, and a description of the training was included in the text.

In lines 136 to 138, the following sentence has been included:

For their sensory training three sessions of 40 min were carried out to identify descriptors in taste to familiarize them with the terminology and use of scales.

Comments 4.

The results indicate a decrease in pH with agavin addition, especially in Q. eduardii formulations. However, the manuscript does not fully explain the underlying mechanisms. Discuss potential reasons for this effect, such as the release of organic acids from agavins or interactions with polyphenols.

R = The requested elements have been incorporated into the discussion section, reorganizing the information from line 679 to line 698. The discussion is as follows:

The pH range of oak infusions is typically between 5.0 and 6.0 (6, 32, 33). In the present study, the initial pH values of the unsweetened Q. sideroxyla and Q. eduardii infusions were 5.42 and 5.84, respectively. The influence of agavins on the pH of Q. sideroxyla was found to be significant (p < 0.001), though the overall decrease (up to 0.2 points at 10%) remained modest. In contrast to the reduction in pH and increase in acidity commonly observed in commercially prepared infusions (34), no correlations were detected in these beverages between pH and the significant increases in acidity in Q. sideroxyla and Q. eduardii infusions resulting from the addition of agavins (see Table 2). A notable observation in the formulation of Q. eduardii was the decrease in pH that occurred with the addition of 2% agavins (5.84 to 5.33), followed by an increase with the addition of 6% (5.70) and a decrease with the addition of 10% (5.62). This phenomenon may be attributed to acid-base interactions between the agavins, and the polyphenols present in the oak leaves. This phenomenon could be attributed to the presence of functional groups capable of forming hydrogen bonds between the hydroxyl groups of the polysaccharides and the carboxyl groups of the phenolic acids, such as hydroxybenzoic acids. The fructan structure of agavins has been hypothesized to enhance the solubility and dissociation of phenolic acids, thereby affecting pH levels. The observed non-linear response may be attributable to competitive interactions that vary with concentration. The polyphenolic profile of Q. eduardii and Q. sideroxyla could also play a role, suggesting the need for further research.

Comments 5.

The study suggests that agavins form hydrogen bonds with phenolic acids, potentially affecting their bioavailability. However, it is unclear whether this interaction enhances or reduces their functional properties. Discuss how these interactions might affect the absorption and health benefits of phenolic compounds in the human body.

R = We appreciate your comment and, in accordance with your suggestion, we have expanded the discussion from line 725 to line 738 to address how the interactions between agavins and phenolic acids could influence the absorption and health benefits of these compounds.

This study hypothesizes that the formation of complexes between polysaccharides and phenolic compounds can influence their bioavailability by modifying their solubility, stability, and absorption in the gastrointestinal tract. The potential for the formation of hydrogen bonds between the hydroxyl and carboxyl groups of phenolic acids, such as quinic acid and hydroxybenzoic acids, and the O-H groups of agavins is of particular interest. In some systems, this interaction with polysaccharides can protect phenolic compounds from oxidative or enzymatic degradation, which could favor their stability and extend their availability in the body. This is particularly relevant in food matrices, where encapsulation in polysaccharide networks can improve absorption in the small intestine. Conversely, the formation of supramolecular complexes can limit the release of free phenols, reducing their solubility in aqueous media and potentially affecting their absorption in the intestine. Depending on the strength of the interaction and the capacity of the intestinal microbiome to release these compounds, this association could diminish their biological effectiveness. In summary, …

Comments 6.

The conclusion mentions the potential health benefits of agavin-sweetened oak iced tea, but the discussion does not explore this in depth. Elaborate on the potential implications for consumer health, referencing existing literature on the metabolic effects of agavins.

R = We appreciate your observation and have expanded the discussion on the potential health benefits of agavin sweetened iced tea, incorporating relevant reference on its metabolic effects.

The following text has been included in the discussion section from line 744 to line 751:

Furthermore, the literature on the effects of prebiotic fibers, such as inulin, on hormones related to appetite and subjective appetite has yielded equivocal results. Studies carried out by Birkeland et al. (2021) have shown that fructans and other prebiotic fibers have the ability to stimulate the secretion of intestinal hormones, such as glucagon-like peptide-1 (GLP-1) and peptide YY (PYY). These hormones play a crucial role in appetite regulation and glycemic control. Consequently, regular consumption of the aforementioned formulations can contribute to the regulation of energy metabolism and the feeling of fullness.

Comments 7.

While the study is comprehensive, it does not discuss potential limitations, such as sample size constraints in sensory evaluations or possible variations in oak leaf composition due to environmental factors. Acknowledge these limitations and suggest future research directions.

R = We acknowledge these limitations and appreciate your valuable suggestions for future research.

Reviewer 4 Report

Comments and Suggestions for Authors

This work investigated the effects of agavin addition on the sensory property, phenolic compounds and antioxidant activity of infusions of Quercus sideroxyla and Quercus eduardii leaves. The topic is interesting with some novelty, and the results may be of some help for the development of functional drink with addition of agavins. The manuscript was well organized, and the results were presented logically. I have some minor concerns that are listed below.

  1. Line 309: ‘agavins are an effective sweetener’ need to be revised.
  2. Figure 1: what does the (a) and (b) represent? In addition, no different letters can be found in the figure.
  3. As indicated in Table 1, addition with 2% agavins in eduardii samples resulted in a significant decrease of pH (0.51), but the pH was increased by 0.37 when the addition of agavins was increased to 6%. However, the pH showed little change in Q. sideroxyla samples. What is the reason?
  4. Table 1: The TTA of Q. sideroxyla samples with addition of 0, 2%, and 6% agavins showed the same value, but they have significant difference. Please check and confirm the data.
  5. The ACE inhibition seems to be unnecessary since little information could be found in the text.
  6. This work highlighted the interaction of agavins with phenolic compounds and the effects on antioxidant activities. However, the effects of different concentration of agavins were not fully discussed.

Author Response

This work investigated the effects of agavin addition on the sensory property, phenolic compounds and antioxidant activity of infusions of Quercus sideroxyla and Quercus eduardii leaves. The topic is interesting with some novelty, and the results may be of some help for the development of functional drink with addition of agavins. The manuscript was well organized, and the results were presented logically. I have some minor concerns that are listed below.

Comments 1.

Line 309: ‘agavins are an effective sweetener’ need to be revised.

R = It is important to highlight that the manuscript refers to the fact that the participating panelists perceived an acceptable sweetness of the agavins, which suggests that agavins may be a good sweetener; further studies will be necessary.

Comments 2.

Figure 1: what does the (a) and (b) represent? In addition, no different letters can be found in the figure.

R = The authors neglected to include this information. The data have been incorporated into the figure caption, as outlined below:

Figure 1. Effect of agavins concentration on sensory acceptability of oak iced tea. (a) Quercus sideroxyla and (b) Quercus eduardii. Data are expressed as mean S.E.M. * indicates statistical difference (n = 24, LSD, p<0.05).

Comments 3.

As indicated in Table 1, addition with 2% agavins in eduardii samples resulted in a significant decrease of pH (0.51), but the pH was increased by 0.37 when the addition of agavins was increased to 6%. However, the pH showed little change in Q. sideroxyla samples. What is the reason?

R = The fluctuations recorded in the pH of the Q. eduardii samples with different concentrations of agavins can be attributed to acid-base interactions between the agavins and the polyphenols present in the oak leaves. As previously mentioned, the initial decrease in pH with 2% agavins (from 5.84 to 5.33), followed by an increase at 6% (from 5.70) and another decrease at 10% (from 5.62), suggests a non-linear response, likely due to competitive interactions between these compounds. The fructan structure of agavins has been postulated to improve the solubility and dissociation of phenolic acids, which can influence pH levels in a concentration-dependent manner.

In contrast, the pH of the Q. sideroxyla samples exhibited modest changes with the addition of agavins, despite the statistical significance of the effect. This discrepancy could be attributed to variations in the polyphenolic composition of Q. sideroxyla compared to Q. eduardii. The hypothesis is reinforced by the fact that the different polyphenolic profiles give rise to different interactions with the agavins, which affects the degree of pH modulation. Consequently, further research is necessary to thoroughly elucidate these effects, which involves investigating the specific polyphenolic composition of both species and their interactions with agavins.

Comments 4.

Table 1: The TTA of Q. sideroxyla samples with addition of 0, 2%, and 6% agavins showed the same value, but they have significant difference. Please check and confirm the data.

R = The observation is appreciated but following a thorough review, it has been determined that the results are accurate. The statistical analysis indicates statistical significance for both Q. sideroxyla and Q. eduardii. The data were analyzed using a one-way analysis of variance (ANOVA) followed by a least significant difference (LSD) post hoc test.

Comments 5.

The ACE inhibition seems to be unnecessary since little information could be found in the text.

R = The observation regarding ACE inhibition is appreciated; its inclusion in the study is relevant due to the fact that aqueous extracts of Q. sideroxyla and Q. eduardii have been previously reported as potent ACE inhibitors. The results obtained in this study indicate that the addition of agavins further enhances the ACE inhibitory effect, suggesting the existence of possible synergistic interactions between agavins and the bioactive compounds present in oak infusions. While the manuscript's primary focus has been on the antioxidant response, given the occurrence of agavins in this context, the observed increase in ACE inhibition with agavins is a noteworthy finding that merits further exploration. In light of the comments received, the discussion in the manuscript has been expanded to better contextualize and emphasize the importance of this effect. The following has been integrated into line 773: 

These interactions have led to an increase in the capacity to inhibit angiotensin-converting enzyme. Despite the fact that there is little information in the literature on the impact of these synergistic interactions, Lee et al. (2015) reported that arginine-fructose (Arg-Fru) enhances ACE inhibition, contributing to antihypertensive effects. Similarly, our study suggests that agavins, which contain fructose-based polysaccharides, may modulate ACE inhibition in oak infusions, potentially through interactions that enhance bioactive compound availability.

Comments 6.

This work highlighted the interaction of agavins with phenolic compounds and the effects on antioxidant activities. However, the effects of different concentration of agavins were not fully discussed.

R = We appreciate your feedback regarding the impact of different concentrations of agavins on the antioxidant response. Our results demonstrate that agavins modulate antioxidant activity in a non-linear way, as described by cubic models (Table 5). These models underscore the concentration-dependent interactions with phenolic compounds, in which both positive and negative effects were observed depending on the species (Q. sideroxyla or Q. eduardii).

Conclusive evidence from FT-IR, UPLC-PDA-ESI-MS/MS, and PLS-DA analyses confirms that these changes are related to modifications in the stability and structure of phenolic compounds. Variations in the antioxidant response in different assays (ABTS, FRAP, ORAC) further illustrate the complex role of agavins in these formulations.

To address your inquiry, I propose expanding the discussion to explicitly highlight these concentration-dependent effects and their implications for the modulation of antioxidant activity, as follows:

Furthermore, the findings indicate that the incorporation of agavins appears to regulate the antioxidant response through multifaceted interactions with the phenolic compounds present in these beverages. Despite a paucity of specific studies on these interactions in beverages, Medina-Larqué et al. (2022) reported that the amalgamation of blueberry polyphenols and agavins in a murine model influenced the bioavailability and biological activity of these compounds. This finding suggests that agavins can affect the stability and functionality of polyphenols in different systems, which could explain the non-linear responses observed in the antioxidant activity of the formulations analyzed. However, further studies are needed to clarify the specific mechanisms of these interactions in food matrices.

In summary, …

Round 2

Reviewer 1 Report

Comments and Suggestions for Authors

The authors have improved the manuscript and corrected the suggestions highlighted in the revision.